# Knee-Deep in C-RASP:
# A Transformer Depth Hierarchy

**Andy Yang**
University of Notre Dame
ayang4@nd.edu

**Michaël Cadilhac**
DePaul University
michael@cadilhac.name

**David Chiang**
University of Notre Dame
dchiang@nd.edu

## Abstract

It has been observed that transformers with greater depth (that is, more layers) have more capabilities, but can we establish formally which capabilities are gained? We answer this question with a theoretical proof followed by an empirical study. First, we consider transformers that round to fixed precision except inside attention. We show that this subclass of transformers is expressively equivalent to the programming language C-RASP and this equivalence preserves depth. Second, we prove that deeper C-RASP programs are more expressive than shallower C-RASP programs, implying that deeper transformers are more expressive than shallower transformers (within the subclass mentioned above). The same is also proven for transformers with positional encodings (like RoPE and ALiBi). These results are established by studying a temporal logic with counting operators equivalent to C-RASP. Finally, we provide empirical evidence that our theory predicts the depth required for transformers without positional encodings to length-generalize on a family of sequential dependency tasks.

## 1 Introduction

Transformers in practice have been getting deeper and deeper over time. The original implementation (Vaswani et al., 2017) used transformers with 8 layers. BERT-Large (Devlin et al., 2019) had 24; GPT-2-XL (Radford et al., 2019) had 48; GPT-3 175B (Brown et al., 2020) had 96. Can we explain what are the effects of deepening the networks?

Figure 1: C-RASP is expressively equivalent to fixed-precision transformers at each depth level. Then, because deeper C-RASP programs can solve more problems, deeper fixed-precision transformers can solve more problems as well.

**Accuracy**

| depth → | 1 | 2 | 3 | 4 | 5 | 6 | 7 | 8 | 9 | 10 |
|---|---|---|---|---|---|---|---|---|---|---|
| $L_3$ | 100 | 100 | 100 | 100 | 100 | 100 | 100 | 100 | 100 | 100 |
| $L_4$ | 22 | 100 | 100 | 100 | 100 | 100 | 100 | 100 | 100 | 100 |
| $L_5$ | 11 | 51 | 100 | 100 | 100 | 100 | 100 | 100 | 100 | 100 |
| $L_6$ | 10 | 8 | 37 | 100 | 98 | 100 | 100 | 100 | 100 | 100 |
| $L_7$ | 8 | 8 | 20 | 49 | 100 | 100 | 100 | 100 | 98 | 100 |
| $L_8$ | 5 | 6 | 19 | 52 | 75 | 95 | 100 | 100 | 100 | 100 |
| $L_9$ | 7 | 7 | 6 | 24 | 29 | 66 | 95 | 99 | 98 | 93 |
| $L_{10}$ | 2 | 2 | 2 | 6 | 12 | 49 | 77 | 96 | 96 | 98 |
| $L_{11}$ | 2 | 2 | 3 | 3 | 6 | 11 | 46 | 31 | 96 | 89 |
| $L_{12}$ | 0 | 0 | 0 | 1 | 2 | 4 | 21 | 30 | 27 | 62 |

Figure 2: Our theoretical results predict that a transformer with depth $k$ can recognize language $L_{k+2}$ but not $L_{k+3}$ (demarcated by the black line). This closely aligns with our experimental results.

39th Conference on Neural Information Processing Systems (NeurIPS 2025).

We can empirically observe capabilities that deeper transformers exhibit which shallower transformers do not. For instance, Clark et al. (2019) and Tenney et al. (2019) find that attention heads at lower layers exhibit lower-order patterns (e.g., each symbol attends to the previous symbol), while heads at higher layers exhibit higher-order patterns (e.g., each direct object attends to its verb). However, we do not have many theoretical guarantees about the impacts of depth in transformers.

In classical theoretical computer science, one studies the power of computational models by asking what *languages* they can express – an equivalent way of asking what *problems* they can solve. Our question becomes: What languages can be expressed by transformers of various depths? We explore this question using the temporal logic TL[$\overleftarrow{\#}$], which is equivalent to the programming language C-RASP (Yang and Chiang, 2024), a variant of the RASP language (Weiss et al., 2021). Previously, C-RASP had been shown to be no less expressive than fixed-precision transformers and no more expressive than arbitrary-precision transformers. In this paper, we prove that when transformers are defined with rounding to fixed precision except inside attention (see Section 2.1 for a more precise statement), **transformers are expressively equivalent to** TL[$\overleftarrow{\#}$]. Moreover, the equivalences between C-RASP, TL[$\overleftarrow{\#}$], and transformers preserve depth.

We can therefore investigate transformer depth by investigating TL[$\overleftarrow{\#}$] depth. Here, we prove a strict depth hierarchy for TL[$\overleftarrow{\#}$], meaning that there is a problem that is solvable by a depth-$k$ TL[$\overleftarrow{\#}$] formula, but not solvable by any depth-$(k-1)$ formulas. This implies **a strict depth hierarchy for C-RASP and transformers** (Fig. 1). (We also prove a strict depth hierarchy for the more expressive logic TL[$\overleftarrow{\#}, \overrightarrow{\#}$]. This implies a strict depth hierarchy for FO[$<$]-uniform LTC$^0$, which was not previously known.) We find experimentally that **the C-RASP depth hierarchy closely predicts** the depth that transformers require to solve problems with particular sequential dependencies (Fig. 2).

The languages $L_k$, which separate transformers of different depths, are just sets of strings with $k$ runs of symbols. This suggests that, for example, in a speech recognition system where each phoneme can extend over multiple frames, a transformer with fixed depth may have difficulty recognizing $k$-grams of phonemes, for $k$ sufficiently large.

We are aware of three previous depth separation results for transformers. Yang et al. (2024) established a strict depth hierarchy for *unique-hard attention* transformers, based on the Until hierarchy for linear temporal logic (Etessami and Wilke, 2000). However, unique-hard attention transformers appear to diverge from transformers as used in practice (Huang et al., 2025; Liu et al., 2023). Sanford et al. (2024) proved, conditioned on a widely known conjecture (Ghaffari et al., 2019), that depth $\Theta(\log(k))$ is necessary and sufficient for transformers to solve the $k$-hop induction heads problem. The first unconditional depth–width tradeoff, based on communication complexity lower bounds, was shown by Chen et al. (2025). In essence, a transformer with depth $k$ would require an impractically large width of $\Omega(\text{poly}(n))$ to perform the sequential composition of $k+1$ functions, while a transformer with depth $k+1$ can implement a solution with a modest width of $O(\text{polylog}(n))$.

The latter two results used transformers whose parameters depended on the sequence length $n$. Our results are *parameter-uniform*, that is, they construct transformers with parameters independent of $n$, making them applicable to inputs of arbitrary length and better predictors of length generalization. Below, we compare and contrast these depth separation results with ours:

| | model | transformer | unconditional | parameter-uniform |
|---|---|---|---|---|
| **Yang et al. (2024)** | temporal logic | unique-hard | **yes** | **yes** |
| **Chen et al. (2025)** | communication complexity | **softmax** | yes | no |
| **Sanford et al. (2024)** | massively parallel computation | **softmax** | no | no |
| **This work** | temporal logic | **softmax** | yes | yes |

## 2 Preliminaries

We write $\mathbb{N}$ for the set of nonnegative integers and $[n]$ for the set $\{1, \ldots, n\}$. With $w = w_1 \cdots w_n$ a string over $\Sigma$, we write $w[i:j]$ for the substring $w_i \cdots w_j$. With $L$ a language, we write $L^*$ for the Kleene star of $L$ ($L^* = \bigcup_{k \geq 0} L^k$) and $L^+$ for $LL^*$. See Appendix G for an index of all notation used.

## 2.1 Transformers

In this paper, we consider transformers that use fixed-precision numbers. Merrill and Sabharwal (2023) have pointed out that in a transformer that uses fixed precision for all operations, there is some length beyond which the attention weights necessarily round to zero, making attention unable to attend to anything. To get around this, we define fixed-precision transformers without rounding of numbers that scale as $n$ or $1/n$. Namely, define an operation on two $n$-dimensional vectors,

$$\text{sumdiv}(\mathbf{a}, \mathbf{b}) = \frac{\sum_{j=1}^{n} \mathbf{a}_j}{\sum_{j=1}^{n} \mathbf{b}_j}.$$

Then if $\mathbf{q} \in \mathbb{R}^{1 \times d}$ is a query vector, $\mathbf{K} \in \mathbb{R}^{n \times d}$ is a matrix of keys, and $\mathbf{v} \in \mathbb{R}^{n \times 1}$ is a value vector, attention can be written as

$$\text{att}(\mathbf{q}, \mathbf{K}, \mathbf{v}) = \text{sumdiv}\big((\exp \mathbf{q}\mathbf{K}^\top) \circ \mathbf{v}, \exp \mathbf{q}\mathbf{K}^\top\big)$$

where $\circ$ is componentwise multiplication. Because the intermediate results of sumdiv scale with $n$, we do not round them; we only round the final result. Our transformers use future-masking but no position encodings. See Appendix B.1 for a full definition.

## 2.2 Temporal logics with counting

Temporal logics are used to express properties of (finite and infinite) strings, using predicates and temporal operates to assert properties of the current position. For instance, in linear temporal logic (Gabbay et al., 1980), one can express things like "at some time in the future, there is an $a$." Temporal logic with counting (Hirshfeld and Rabinovich, 2012; Barceló et al., 2024) adds more general integer-valued counting operators: a property such as "at some time in the future, there is an $a$" thus becomes "the number of $a$'s in the future is at least 1." Here, we are interested in these logics because they are equivalent to (variants of) transformers (Section 3 and Appendix B).

**Definition 2.1.** *The syntax of* past temporal logic with counting, *or* $\mathsf{TL}[\overleftarrow{\#}]$, *is as follows:*

$$
\begin{aligned}
\phi &::= Q_\sigma \mid t_1 < t_2 \mid \neg\phi_1 \mid \phi_1 \wedge \phi_2 & \sigma \in \Sigma & \qquad \textit{Boolean-valued formulas} \\
t &::= \overleftarrow{\#}[\phi_1] \mid t_1 + t_2 \mid 1 & & \qquad \textit{integer-valued terms}
\end{aligned}
$$

Temporal logic with counting, or $\mathsf{TL}[\overleftarrow{\#}, \overrightarrow{\#}]$, adds an operator $\overrightarrow{\#}$, and is discussed in Appendix D. In Appendix F.1, we further extend the logic with a MOD predicate and $\mathbf{Y}$ operator corresponding to positional encodings in transformers. Other operators, like $\leq, \geq, \vee, -$, multiplication by integer constants, and integer constants other than 1, can be defined in terms of the above.

The semantics of $\mathsf{TL}[\overleftarrow{\#}]$ and $\mathsf{TL}[\overleftarrow{\#}, \overrightarrow{\#}]$ are defined by the relation $w, i \models \phi$, meaning that the formula $\phi$ is satisfied by string $w$ at position $i$. First, $w, i \models Q_\sigma$ holds when the $i$-th symbol of $w$ is $\sigma$. The term $\overleftarrow{\#}[\phi]$, evaluated on string $w$ at position $i$, counts the number of positions in $w[1:i]$ that satisfy $\phi$. Similarly, $\overrightarrow{\#}[\phi]$ counts the number of positions in $w[i:n]$ that satisfy $\phi$ (where $n = |w|$). Terms can be added (+) or compared (<). See Definition A.1 for the full definition. We write $w \models \phi$ iff $w, |n| \models \phi$; that is, $\phi$ is satisfied by $w$ at its *final* position.[1] Finally, the language defined by $\phi$ $\mathcal{L}(\phi) = \{w \in \Sigma^* \mid w \models \phi\}$.

**Example 2.2.** *The Dyck language is the set of strings of balanced and matched parentheses. The formula $\phi_{balance} = (\overleftarrow{\#}[Q_(] = \overleftarrow{\#}[Q_)])$ checks that the number of left and right parentheses is equal. The formula $\phi_{match} = (\overleftarrow{\#}[\overleftarrow{\#}[Q_(] < \overleftarrow{\#}[Q_)]] = 0)$ checks that, in every prefix, the number of right parentheses does not exceed the number of left parentheses. So the Dyck language is defined by $\phi_{balance} \wedge \phi_{match}$. Appendix A.2 shows more detailed traces of this formula on some example strings.*

The *depth* of a formula or term of $\mathsf{TL}[\overleftarrow{\#}]$ or $\mathsf{TL}[\overleftarrow{\#}, \overrightarrow{\#}]$ is the maximum depth to which $\overleftarrow{\#}$ and $\overrightarrow{\#}$ operators are nested. $\mathsf{TL}[\overleftarrow{\#}]_k$ is the class of all $\mathsf{TL}[\overleftarrow{\#}]$ formulas with depth at most $k$, and similarly for $\mathsf{TL}[\overleftarrow{\#}, \overrightarrow{\#}]_k$ and the other logics we use. See Definition A.2 in Appendix A.1 for a formal definition. A depth-1 formula may be a Boolean combination of other formulas of depth 0 and 1; a *minimal depth-1 formula* is one that does not contain other depth-1 formulas, that is, a formula of the form $t_1 < t_2$

---

[1]Usually, the semantics of temporal logics is defined by $w \models \phi$ iff $w, 1 \models \phi$. However, in our setting, our definition mimics the behavior of generative transformer decoders, which for each position, consider all previously generated tokens (to the *left*) in order to produce the next token.

where $t_1$ and $t_2$ are depth-1 terms. For example, $Q_a \wedge (1 \leq 1+1)$ has depth 0, $Q_a \wedge (\overleftarrow{\#}[Q_a] < \overleftarrow{\#}[Q_b])$ has depth 1 but is not minimal, and $\overleftarrow{\#}[Q_a] \leq \overleftarrow{\#}[Q_b] + \overleftarrow{\#}[Q_c] + 1$ is a minimal depth-1 formula.

Yang and Chiang (2024) called $\mathsf{TL}[\overleftarrow{\#}]$ by a different name, $\mathsf{K_t}[\#]$. They showed that it is equivalent to C-RASP, which admits a notion of depth that corresponds exactly to formula depth.

## 2.3 Parikh vectors

At the heart of $\mathsf{TL}[\overleftarrow{\#}]$ is the ability to count symbols, so we introduce some notation related to counts.

**Definition 2.3** (Parikh, 1966). *Let $\Sigma = \{\sigma_1, \sigma_2, \ldots, \sigma_m\}$ be an (ordered) alphabet. The* Parikh vector *of $w$, written $\Psi(w)$, records the number of times each symbol occurs in $w$, ignoring order. That is,*

$$\Psi \colon \Sigma^* \to \mathbb{N}^m$$
$$\Psi_\sigma(w) = |\{i \in [|w|] \mid w_i = \sigma\}|$$
$$\Psi(w) = (\Psi_{\sigma_1}(w), \Psi_{\sigma_2}(w), \ldots, \Psi_{\sigma_m}(w)).$$

**Definition 2.4.** *For a Parikh vector $\vec{v} = (v_1, v_2, \ldots, v_{|\Sigma|}) \in \mathbb{N}^{|\Sigma|}$ we write the length of $\vec{v}$ as $\|\vec{v}\| = v_1 + v_2 + \ldots + v_{|\Sigma|}$. To access individual coordinates, we write $[\vec{v}]_{\sigma_i}$ for $v_i$.*

**Definition 2.5.** *For Parikh vectors $\vec{i}, \vec{j} \in \mathbb{N}^{|\Sigma|}$, we write $\vec{i} \leq \vec{j}$ iff $[\vec{i}]_\sigma \leq [\vec{j}]_\sigma$ for all $\sigma \in \Sigma$. We write $[\vec{i}, \vec{j}]$ for the set of all vectors $\vec{v}$ such that $\vec{i} \leq \vec{v} \leq \vec{j}$. We call $[\vec{i}, \vec{j}]$ an* interval *in $\mathbb{N}^{|\Sigma|}$, and we write $\mathcal{I}(\mathbb{N}^{|\Sigma|})$ for the set of all intervals in $\mathbb{N}^{|\Sigma|}$. A family of intervals is a function $I \colon \mathbb{N}^{|\Sigma|} \to \mathcal{I}(\mathbb{N}^{|\Sigma|})$.*

**Definition 2.6.** *For each partial function $\pi \colon \mathbb{N}^{|\Sigma|} \times \mathbb{N} \to \{0, 1\}$ such that $\pi(\vec{v}, i)$ is defined iff $1 \leq i \leq \|\vec{v}\|$, define a predicate $\Pi$, called a* Parikh numerical predicate *or PNP:*

$$w, i \models \Pi \iff \pi(\Psi(w), i) = 1.$$

*We write $\mathsf{TL}[\overleftarrow{\#}, \mathsf{PNP}]$ and $\mathsf{TL}[\overleftarrow{\#}, \overrightarrow{\#}, \mathsf{PNP}]$ for the logics $\mathsf{TL}[\overleftarrow{\#}]$ and $\mathsf{TL}[\overleftarrow{\#}, \overrightarrow{\#}]$, respectively, augmented with arbitrary PNPs.*

## 2.4 Piecewise testable languages

Piecewise testable languages are a subclass of regular languages that has been studied in semigroup theory and logic (Simon, 1975). Here, they will be key to separating the depth levels of both $\mathsf{TL}[\overleftarrow{\#}]$ and $\mathsf{TL}[\overleftarrow{\#}, \overrightarrow{\#}]$.

**Definition 2.7.** *A $\mathcal{J}$-expression is a language of the form $\Sigma^* \sigma_1 \Sigma^* \sigma_2 \Sigma^* \cdots \Sigma^* \sigma_k \Sigma^*$ where $\sigma_1, \ldots, \sigma_k \in \Sigma$. A language is $k$-*piecewise testable *if it is a Boolean combination of $\mathcal{J}$-expressions with at most $k$ fixed symbols. A language is* piecewise testable *if it is $k$-piecewise testable for some $k$.*

**Lemma 2.8.** *For all $k \geq 0$, define $L_k$ to be the language of strings with alternating blocks of $a$'s and $b$'s, starting with $a$:*

$$L_k = \begin{cases} (a^+ b^+)^{k/2} & k \text{ even} \\ (a^+ b^+)^{(k-1)/2} a^+ & k \text{ odd.} \end{cases} \tag{1}$$

*Then $L_k$ is $k$-piecewise testable.*

*Proof.* Define the following $k$-piecewise testable languages:

$$A_k = \begin{cases} \Sigma^* (a\Sigma^* b\Sigma^*)^k & k \text{ even} \\ \Sigma^* (a\Sigma^* b\Sigma^*)^{(k-1)/2} a\Sigma^* & k \text{ odd} \end{cases} \qquad B_k = \begin{cases} \Sigma^* (b\Sigma^* a\Sigma^*)^{k/2} & k \text{ even} \\ \Sigma^* (b\Sigma^* a\Sigma^*)^{(k-1)/2} b\Sigma^* & k \text{ odd.} \end{cases} \tag{2}$$

Then $L_k$ is a Boolean combination of these:

$$L_k = \Sigma^* \setminus B_k \cap A_k. \qquad \square$$

**Lemma 2.9.** *Any $k$-piecewise testable language is definable in $\mathsf{TL}[\overleftarrow{\#}]_k$, and any $(2k + 1)$-piecewise testable language is definable in $\mathsf{TL}[\overleftarrow{\#}, \overrightarrow{\#}]_{k+1}$.*

*Proof sketch.* Since our logics are closed under Boolean operations, we simply need to show the statement for $\mathcal{J}$-expressions. Let us sketch this in the case $k = 1$. There is a straightforward way to define $\Sigma^* a\Sigma^* b\Sigma^* c\Sigma^*$ in $\mathsf{TL}[\overleftarrow{\#}]_{2k+1}$:

$$\overleftarrow{\#}[(\overleftarrow{\#}[(\overleftarrow{\#}[Q_a] \geq 1) \wedge Q_b] \geq 1) \wedge Q_c] \geq 1.$$

With both past and future counting, we can find the middle symbol $b$ and check to the left and right:

$$\overleftarrow{\#}[\overleftarrow{\#}[Q_a] \geq 1 \wedge Q_b \wedge \overrightarrow{\#}[Q_c] \geq 1] \geq 1.$$

This is in $\mathsf{TL}[\overleftarrow{\#}, \overrightarrow{\#}]_{k+1}$, as desired. See Appendix A.4 for the full proof. $\square$

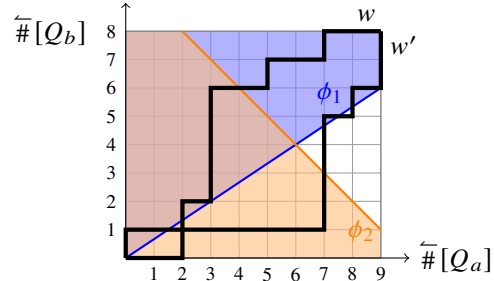
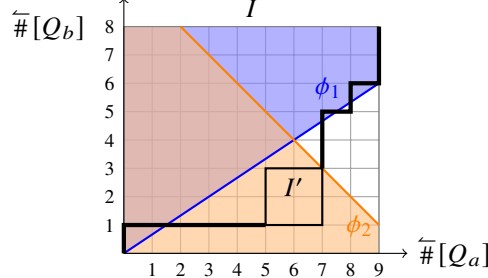

(a) Strings can be pictured as paths, and minimal depth-1 subformulas as half-planes.

(b) Within a sufficiently large rectangle ($I$), one can find a sub-rectangle ($I'$) within which the depth-1 sub-formulas are either always true or always false.

Figure 3: Visualization of the reduction lemma (Lemma 4.9).

## 3 Transformer Equivalence

Logics like $\mathsf{TL}[\overleftarrow{\#}]$ provide a way to reason about the computations that occur in transformers. Yang and Chiang (2024) proved that transformers can simulate $\mathsf{TL}[\overleftarrow{\#}]$, and $\mathsf{TL}[\overleftarrow{\#}]$ can simulate transformers that round all values to fixed precision. Here, we show that fixed precision, with rounding slightly loosened as in Section 2.1, makes it possible to obtain an exact equivalence.

**Theorem 3.1.** *A language $L$ is defined by a formula of* $\mathsf{TL}[\overleftarrow{\#}]$ *of depth $k$ if and only if* <BOS> $\cdot L$ *is recognized by a fixed-precision transformer of depth $k$.*

*Proof.* See Appendix B. □

In Appendix F, we will extend the transformer-to-logic direction of this result to several position encodings and an extension of $\mathsf{TL}[\overleftarrow{\#}]$.

## 4 Depth Hierarchy

To prove a strict depth hierarchy for $\mathsf{TL}[\overleftarrow{\#}]$, we adapt the technique of Behle et al. (2009), which was originally used on $\widehat{\mathsf{MAJ}}_2[<]$, a logic equivalent to $\mathsf{TL}[\overleftarrow{\#}, \overrightarrow{\#}]$. The main idea is to assume, towards a contradiction, that a certain language *is* definable by a formula, then to simultaneously restrict the language and reduce the depth of the formula down to depth 1. The contradiction will be that the restricted language has a property (namely, sensitivity to ordering) that *is not* definable at depth 1. This technique can also be applied to the bidirectional logic $\mathsf{TL}[\overleftarrow{\#}, \overrightarrow{\#}]$, with implications for other logics and circuit classes, as detailed in Appendix D.

### 4.1 Intuition and example

As an example of the technique, consider the following formula:

$$\phi = \overleftarrow{\#}[\underbrace{(2\overleftarrow{\#}[Q_a] \le 3\overleftarrow{\#}[Q_b])}_{\phi_1} \wedge \underbrace{(\overleftarrow{\#}[Q_a] + \overleftarrow{\#}[Q_b] \le 10)}_{\phi_2} \wedge Q_b] \ge 1.$$

This formula has depth 2 and it has two minimal depth-1 subformulas, $\phi_1$ and $\phi_2$. We want to replace $\phi_1$ and $\phi_2$ with depth-0 subformulas, reducing the depth of $\phi$ from 2 to 1, while restricting the language defined.

Since $\phi_1$ and $\phi_2$ are linear inequalities in the counts $\overleftarrow{\#}[Q_\sigma]$, we can picture them as half-planes (Fig. 3a). A string can be pictured as a path, and a prefix of the string as a point on the path. For concreteness, we fix a vector $\vec{n} = (9, 8)$ and let $I$ be the interval $[\vec{0}, \vec{n}]$. (In the formal proof, $\vec{n}$ will not be fixed, and $I$ will be a *family* of intervals depending on $\vec{n}$.) Suppose we can find some subinterval $I'$ (which can be pictured as a rectangle, as in Fig. 3b) that does not cross any of the half-plane boundaries. All points in $I'$ are equivalent in the sense that the truth value of $\phi_1$ and $\phi_2$ is the same for all points in $I'$.

We restrict the language by choosing a prefix corresponding to a path from the bottom left of $I$ to the bottom left of $I'$, and a suffix corresponding to a path from the top right of $I'$ to the top right of $I$. In our example, we picked *baaaaa* and *bbababb*. Informally, we can define the restriction of $\phi$ to this prefix and suffix like so:

$$\phi' = \text{``prefix is } baaaaa\text{''} \wedge \text{``suffix is } bbababb\text{''}$$
$$\wedge \; (\overleftarrow{\#}[\underbrace{\phi_1 \wedge \phi_2 \wedge Q_b \wedge \text{``inside } I'\text{''}}_{(*)}] + \overleftarrow{\#}[\underbrace{\phi_1 \wedge \phi_2 \wedge Q_b \wedge \text{``in prefix/suffix''}}_{(\dagger)}] \geq 1).$$

Then the occurrences of $\phi_1$ and $\phi_2$ marked $(*)$ are always true and false, respectively, while the occurrences marked $(\dagger)$ depend only on the position. As we will see, we can replace all of these by PNPs, reducing the depth of the formula from 2 to 1.

Carefully iterating this process (Lemma 4.9) leads to a depth-1 formula defining a restriction of the original language. In this language, we will show that the ordering of symbols matters, but we will see in Lemma 4.6 that depth-1 formulas are (in a particular sense) insensitive to the ordering of symbols. This is a contradiction, demonstrating that the original language is not definable.

## 4.2 Affix restrictions

We start by defining the affix restrictions that were informally introduced in at the start of this section. Krebs (2008) enforced similar restrictions using the algebraic tool of *non-uniform morphisms*, while Behle et al. (2009) used numerical predicates on languages with a restricted Parikh image. We follow the latter approach of expressing this idea within a purely logical framework, but introduce Parikh numerical predicates as a way to generalize the technique.

**Definition 4.1** (cf. Def. 6.5 of Krebs, 2008)**.** *An* affix restriction *is a pair* $(\lambda, \varrho)$*, where* $\lambda, \varrho \colon \mathbb{N}^{|\Sigma|} \to \Sigma^*$*. For any language* $L \subseteq \Sigma^*$ *and affix restriction* $(\lambda, \varrho)$*, we define the* restriction *of* $L$ *to* $(\lambda, \varrho)$ *as*

$$_\lambda L_\varrho = \{w \in L \colon \exists w' \in \Sigma^* \text{ such that } w = \lambda(\Psi(w))\, w'\, \varrho(\Psi(w))\}.$$

In the definition above, the set of positions occupied by $w'$ within each string is the only part not fixed by $(\lambda, \varrho)$. This region is important for the depth reduction process, so we give it a name:

**Definition 4.2.** *The* middle *of an affix restriction* $(\lambda, \varrho)$ *is the family of intervals given by* $\vec{n} \mapsto [\Psi(\lambda(\vec{n})), \vec{n} - \Psi(\varrho(\vec{n}))]$.

In order for the middle of an affix restriction to not be too restricted, we typically require affix restrictions to have the following property.

**Definition 4.3.** *We say that a family of intervals* $I$ *is* accommodating *if, for any* $\vec{s} \in \mathbb{N}^{|\Sigma|}$*, there is an interval* $[\vec{i}, \vec{j}]$ *in the image of* $I$ *such that* $\vec{s} \leq \vec{j} - \vec{i}$*, or, equivalently,* $\vec{s} \in [\vec{0}, \vec{j} - \vec{i}]$*. We say that an affix restriction* $(\lambda, \varrho)$ *is* accommodating *if its middle is accommodating.*

An example of a non-accommodating affix restriction, with $\Sigma = \{a, b\}$, is $\lambda((n_a, n_b)) = \epsilon$ and $\varrho((n_a, n_b)) = a^{n_a} b^{n_b}$. In this case $_\lambda L_\varrho$ will have at most one string for each $(n_a, n_b)$. An accommodating affix restriction is the trivial one $\lambda((n_a, n_b)) = \varrho((n_a, n_b)) = \epsilon$. In this case $_\lambda L_\varrho$ will have $\binom{n_a + n_b}{n_a}$ strings for each $(n_a, n_b)$.

Accommodating affix restrictions will be key to depth-reduction. If a language has an accommodating middle under a restriction $(\lambda, \varrho)$, then there is enough room to apply another restriction $(\lambda', \varrho')$ inside the middle, while decreasing the depth of the formula. As long as the property of accommodation is preserved at each step, this process can be iterated until we reach depth 1.

## 4.3 Properties of depth 1

In this section, we exhibit inherent limitations of depth-1 formulas. Informally, we show that these formulas only recognize commutative languages, that is, languages where the order of symbols does not matter. But we need to qualify this statement slightly, and we need some technicalities as well.

In particular, affix restrictions will fix the ordering of symbols in the prefix and suffix, so affix-restricted languages can only be commutative in the following sense:

**Definition 4.4.** *We say that a language $L$ is* commutative on the middle of *an affix restriction $(\lambda, \varrho)$ if, for any $w, w' \in {}_\lambda \Sigma_\varrho^*$ such that $\Psi(w) = \Psi(w')$, we have $w \in L$ if and only if $w' \in L$.*

We will use PNPs to enforce ordering in the prefix and suffix, but to allow languages to be commutative on the middle, we need to prevent the PNPs from enforcing ordering in the middle.

**Definition 4.5.** *We say that a formula $\phi$ is* constant *on a family of intervals $I \colon \mathbb{N}^{|\Sigma|} \to \mathcal{I}(\mathbb{N}^{|\Sigma|})$ if the following holds for all $\vec{n} \in \mathbb{N}^{|\Sigma|}$: For all $w, w' \in \Sigma^*$ with $\Psi(w) = \Psi(w') = \vec{n}$ and all positions $i, i'$ in $I(\vec{n})$, we have $w, i \models \phi$ if and only if $w', i' \models \phi$.*

**Lemma 4.6** (Commutativity of depth 1). *For any depth-$1$ formula $\phi$ of $\mathsf{TL}[\overleftarrow{\#}, \mathsf{PNP}]_1$ or $\mathsf{TL}[\overleftarrow{\#}, \overrightarrow{\#}, \mathsf{PNP}]_1$ and any affix restriction $(\lambda, \varrho)$ with $|\varrho(\vec{n})| \geq 1$ for all $\vec{n}$, if the PNPs of $\phi$ are constant on the middle of $(\lambda, \varrho)$, then $\mathcal{L}(\phi)$ is commutative on the middle of $(\lambda, \varrho)$.*

*Proof.* See Appendix C.1. The reason for the condition $|\varrho(\vec{n})| \geq 1$ is that the $Q_\sigma$ predicates are able to test the symbol in the last position. □

## 4.4 Cropping and reduction lemmas

In this section, we show how to decrease the quantifier depth of a formula $\phi$ while specifying precisely how $\mathcal{L}(\phi)$ is weakened. Our approach follows Lemma 3 of Behle et al. (2009) and Lemma 6.8 of Krebs (2008), but faces additional technical difficulties specific to $\mathsf{TL}[\overleftarrow{\#}]$ and, to a lesser extent, the use of PNPs. Since we will be applying this technique on two-letter alphabets, we set $\Sigma = \{a, b\}$ for the rest of this section. Thus, we can visualize Parikh vectors and intervals in the plane, with the number of $a$'s on the horizontal axis, and the number of $b$'s on the vertical axis.

First, the cropping lemma takes a family of intervals $I$ and "crops" it down to a family of subintervals $I'$ on which the minimal depth-1 subformulas are constant. This is done while controlling where $I'$ sits within $I$, and we start by formalizing this notion:

**Definition 4.7.** *We say that an interval $[\vec{i}', \vec{j}']$* sticks to the top of *an interval $[\vec{i}, \vec{j}]$ if $[\vec{i}', \vec{j}'] \subseteq [\vec{i}, \vec{j}]$ and $[\vec{j}']_b = [\vec{j}]_b$. Graphically, this means that the top edge of the rectangle $[\vec{i}', \vec{j}']$ is included in the top edge of the rectangle $[\vec{i}, \vec{j}]$. We define an interval* sticking to the bottom, left, *or* right *analogously.*

**Lemma 4.8** (Cropping Lemma for $\mathsf{TL}[\overleftarrow{\#}]$). *For any formula $\phi$ of $\mathsf{TL}[\overleftarrow{\#}, \mathsf{PNP}]$ and any accommodating family of intervals $I \colon \mathbb{N}^{|\Sigma|} \to \mathcal{I}(\mathbb{N}^{|\Sigma|})$ such that the PNPs of $\phi$ are constant on $I$, there exists an accommodating family of intervals $I' \colon \mathbb{N}^{|\Sigma|} \to \mathcal{I}(\mathbb{N}^{|\Sigma|})$ such that $I'(\vec{n})$ sticks only to the top (and no other side) of $I(\vec{n})$ for all $\vec{n} \in N^{|\Sigma|}$, and all of the minimal depth-1 subformulas (and PNPs) of $\phi$ are constant on $I'$. Additionally, there exists such an $I'$ such that $I'(\vec{n})$ sticks only to the right of $I(\vec{n})$.*

*Proof.* See Appendix C.2. □

Second, the reduction lemma takes an affix restriction (whose middle is $I'$ given by the cropping lemma) and rewrites away the minimal depth-1 subformulas, reducing the depth of the formula by 1.

**Lemma 4.9** (Reduction Lemma). *For any depth-$k$ formula $\phi$ of $\mathsf{TL}[\overleftarrow{\#}, \mathsf{PNP}]_k$ (or $\mathsf{TL}[\overleftarrow{\#}, \overrightarrow{\#}, \mathsf{PNP}]_k$) and affix restriction $(\lambda, \varrho)$, if the PNPs and minimal depth-1 subformulas of $\phi$ are constant on the middle of $(\lambda, \varrho)$, then there is a formula $\phi'$ of depth $(k-1)$ of $\mathsf{TL}[\overleftarrow{\#}, \mathsf{PNP}]_{k-1}$ (or $\mathsf{TL}[\overleftarrow{\#}, \overrightarrow{\#}, \mathsf{PNP}]_k$, resp.) that defines ${}_\lambda \mathcal{L}(\phi)_\varrho$, and the PNPs of $\phi'$ are constant on the middle of $(\lambda, \varrho)$.*

*Proof.* See Appendix C.3. □

## 4.5 Non-definability results

As described by Behle et al. (2009), the key to applying this lemma is to choose a language $L$ and appropriate affix families such that the restricted language does not become trivial. We now use the cropping and reduction lemmas to derive the strictness of the depth hierarchy of $\mathsf{TL}[\overleftarrow{\#}]$.

**Theorem 4.10.** *Let $k > 0$. The language $L_{k+1}$ (Eq. (1)) is definable in $\mathsf{TL}[\overleftarrow{\#}]_{k+1}$ but not in $\mathsf{TL}[\overleftarrow{\#}]_k$.*

*Proof.* Assume that $k$ is even (the odd case is similar). For a contradiction, assume there exists some depth-$k$ formula $\phi \in \mathsf{TL}[\overleftarrow{\#}]_k$ which defines $L_{k+1}$. Let $I_k(\vec{n}) = [(1, 0), \vec{n} - (1, 0)]$, $\lambda_k(\vec{n}) = a$, $\varrho_k(\vec{n}) = a$, and $\phi_k = \phi$. Note that $(\lambda_k, \varrho_k)$ is accommodating, and $\phi_k$ has no PNPs.

For $\ell = k - 1, k - 2, \ldots, 1$, we will define the following:

1. An accommodating family of intervals $I_\ell(\vec{n}) \subseteq I_{\ell+1}(\vec{n})$;
2. An accommodating affix restriction $\lambda_\ell(\vec{n}) \in L_{k-\ell+1}$ and $\varrho_\ell(\vec{n}) \in a^+$;
3. A depth-$\ell$ formula $\phi_\ell \in \mathsf{TL}[\overleftarrow{\#}, \mathsf{PNP}]_\ell$ which defines $_{\lambda_\ell}(L_{k+1})_{\varrho_\ell}$ and only uses PNPs which are constant over $(\lambda_\ell, \varrho_\ell)$.

We use the following iterative procedure:

1. Using Lemma 4.8, find an accommodating family of intervals $I_\ell$ such that for all $\vec{n}$, $I_\ell(\vec{n})$ sticks only to the top of $I_{\ell+1}(\vec{n})$, and the minimal depth-1 subformulas of $\phi_{\ell+1}$ are constant on $I_\ell(\vec{n})$.
2. Choose an affix restriction whose middle is $I_\ell$, as follows. Let $[\vec{i}, \vec{j}] = I_{\ell+1}(\vec{n})$ and $[\vec{i'}, \vec{j'}] = I_\ell(\vec{n})$; then

$$\lambda_\ell(\vec{n}) = \begin{cases} \lambda_{\ell+1}(\vec{n}) a^{[\vec{i'}-\vec{i}]_a} b^{[\vec{i'}-\vec{i}]_b} & \ell \text{ odd} \\ \lambda_{\ell+1}(\vec{n}) b^{[\vec{i'}-\vec{i}]_b} a^{[\vec{i'}-\vec{i}]_a} & \ell \text{ even} \end{cases}$$

$$\varrho_\ell(\vec{n}) = a^{[\vec{j}-\vec{j'}]_a} \varrho_{\ell+1}(\vec{n}).$$

Because $\lambda_{\ell+1}(\vec{n}) \in L_{k-\ell}$ and $I_\ell(\vec{n})$ sticks only to the top of $I_{\ell+1}(\vec{n})$, we have that $\lambda_\ell \in L_{k-\ell+1}$. At the same time, $\varrho_\ell(\vec{n}) \in a^+$.

3. Using Lemma 4.9, we can find a depth-$\ell$ formula $\phi_\ell$ of $\mathsf{TL}[\overleftarrow{\#}, \mathsf{PNP}]_\ell$ that defines $_{\lambda_\ell}(L_{k+1})_{\varrho_\ell}$ and only uses PNPs which are constant on $(\lambda_\ell, \varrho_\ell)$.

At the end of the procedure above, we are left with the accommodating affix restriction $\lambda_1(\vec{n}) \in L_k$ and $\varrho_1(\vec{n}) \in a^+$, as well as the depth-1 formula $\phi_1 \in \mathsf{TL}[\overleftarrow{\#}, \mathsf{PNP}]_1$, which defines $_{\lambda_1}(L_{k+1})_{\varrho_1}$ and only uses PNPs which are constant over $(\lambda_1, \varrho_1)$.

Since $(\lambda_1, \varrho_1)$ is accommodating, choose $\vec{n}$ so that the middle has $s_a \geq 1$ occurrences of $a$ and $s_b \geq 1$ occurrences of $b$. Construct the strings

$$w = \lambda_1(\vec{n}) b^{s_b} a^{s_a} \varrho_1(\vec{n})$$
$$w' = \lambda_1(\vec{n}) a^{s_a} b^{s_b} \varrho_1(\vec{n}).$$

The prefix $\lambda_1(\vec{n})$ has $k$ blocks ending with $b$, and the suffix $\rho_1(\vec{n})$ is all $a$'s. So $w$ has $(k + 1)$ blocks and is therefore in $L_{k+1}$, while $w'$ has $(k + 3)$ blocks and is therefore not in $L_{k+1}$. But by Lemma 4.6, we have $w \models \phi_1 \iff w' \models \phi_1$. This is a contradiction, so we conclude that no formula $\phi$ with depth $k$ can define $L_{k+1}$.

If $k$ is odd, the argument is the same, with the following changes. First, symbols $a$ and $b$ are swapped, but $\lambda_\ell(\vec{n})$ still starts with $a$. Second, $I_\ell(\vec{n})$ sticks to the right of $I_{\ell+1}(\vec{n})$ instead of the top.

Finally, by Lemma 2.8, $L_{k+1}$ is $(k + 1)$-piecewise testable. Thus, by Lemma 2.9, $L_{k+1}$ is definable in $\mathsf{TL}[\overleftarrow{\#}]_{k+1}$. □

This depth hierarchy on $\mathsf{TL}[\overleftarrow{\#}]$ implies a depth hierarchy for fixed-precision transformers.

**Theorem 4.11.** *A depth-$(k + 1)$ fixed-precision transformer can recognize $L_{k+1}$, but no depth-$k$ fixed-precision transformer can.*

This also implies a depth hierarchy for transformers using several commonly used positional encodings (with a different separating language). Definitions and details can be found in Appendix F.

**Theorem 4.12.** *A depth-$(k + 1)$ fixed-precision transformer can recognize $E_{k+1}$, but no depth-$k$ fixed-precision transformer can, if the transformers can use sinusoidal positional embeddings (Vaswani et al., 2017), RoPE (Su et al., 2024), or ALiBi (Press et al., 2022).*

## 5 Experiments

Our depth hierarchy result suggests that transformers will require greater depth in order to model deeper sequential dependencies. We empirically validate this by training future-masked transformers with no positional encodings and varying depths to learn the $L_k$ language, for varying $k$.[2] Here, the

---

[2]The code used for our experiments is provided at https://github.com/pentagonalize/CRASP_depth. LLMs were used to assist in writing code and debugging.

Accuracy on [201, 250]

| depth → | 1 | 2 | 3 | 4 | 5 | 6 | 7 | 8 | 9 | 10 |
|---|---|---|---|---|---|---|---|---|---|---|
| $L_3$ | 100 | 100 | 100 | 100 | 100 | 100 | 100 | 100 | 100 | 100 |
| $L_4$ | 41 | 100 | 100 | 100 | 100 | 100 | 100 | 100 | 100 | 100 |
| $L_5$ | 36 | 76 | 100 | 100 | 100 | 100 | 100 | 100 | 100 | 100 |
| $L_6$ | 36 | 37 | 74 | 100 | 100 | 100 | 100 | 100 | 100 | 100 |
| $L_7$ | 39 | 40 | 53 | 87 | 100 | 100 | 100 | 100 | 100 | 100 |
| $L_8$ | 37 | 42 | 53 | 78 | 96 | 100 | 100 | 100 | 100 | 100 |
| $L_9$ | 32 | 41 | 45 | 61 | 74 | 93 | 100 | 100 | 100 | 100 |
| $L_{10}$ | 34 | 36 | 39 | 50 | 57 | 80 | 91 | 100 | 100 | 100 |
| $L_{11}$ | 37 | 41 | 45 | 49 | 55 | 60 | 77 | 77 | 100 | 100 |
| $L_{12}$ | 27 | 23 | 34 | 34 | 36 | 47 | 59 | 74 | 73 | 86 |

Accuracy on [251, 300]

| depth → | 1 | 2 | 3 | 4 | 5 | 6 | 7 | 8 | 9 | 10 |
|---|---|---|---|---|---|---|---|---|---|---|
| $L_3$ | 100 | 100 | 100 | 100 | 100 | 100 | 100 | 100 | 100 | 100 |
| $L_4$ | 32 | 100 | 100 | 100 | 100 | 100 | 100 | 100 | 100 | 100 |
| $L_5$ | 24 | 65 | 100 | 100 | 100 | 100 | 100 | 100 | 100 | 100 |
| $L_6$ | 26 | 24 | 59 | 100 | 100 | 100 | 100 | 100 | 100 | 100 |
| $L_7$ | 27 | 24 | 39 | 74 | 100 | 100 | 100 | 100 | 100 | 100 |
| $L_8$ | 23 | 26 | 43 | 82 | 93 | 100 | 100 | 100 | 100 | 100 |
| $L_9$ | 31 | 28 | 25 | 50 | 59 | 91 | 100 | 100 | 100 | 99 |
| $L_{10}$ | 19 | 17 | 17 | 26 | 37 | 71 | 91 | 99 | 100 | 100 |
| $L_{11}$ | 19 | 20 | 21 | 26 | 33 | 42 | 69 | 64 | 100 | 98 |
| $L_{12}$ | 6 | 5 | 8 | 12 | 12 | 21 | 53 | 60 | 56 | 75 |

Accuracy on [301, 350]

| depth → | 1 | 2 | 3 | 4 | 5 | 6 | 7 | 8 | 9 | 10 |
|---|---|---|---|---|---|---|---|---|---|---|
| $L_3$ | 100 | 100 | 100 | 100 | 100 | 100 | 100 | 100 | 100 | 100 |
| $L_4$ | 26 | 100 | 100 | 100 | 100 | 100 | 100 | 100 | 100 | 100 |
| $L_5$ | 15 | 59 | 100 | 100 | 100 | 100 | 100 | 100 | 100 | 100 |
| $L_6$ | 17 | 14 | 49 | 100 | 99 | 100 | 100 | 100 | 100 | 100 |
| $L_7$ | 13 | 12 | 25 | 59 | 100 | 100 | 100 | 100 | 99 | 100 |
| $L_8$ | 11 | 14 | 29 | 68 | 85 | 99 | 100 | 100 | 100 | 100 |
| $L_9$ | 14 | 13 | 11 | 34 | 43 | 80 | 98 | 100 | 99 | 97 |
| $L_{10}$ | 6 | 5 | 5 | 12 | 24 | 58 | 84 | 98 | 98 | 99 |
| $L_{11}$ | 6 | 6 | 6 | 8 | 14 | 21 | 59 | 45 | 99 | 96 |
| $L_{12}$ | 2 | 2 | 2 | 3 | 4 | 9 | 37 | 42 | 38 | 68 |

Accuracy on [351, 400]

| depth → | 1 | 2 | 3 | 4 | 5 | 6 | 7 | 8 | 9 | 10 |
|---|---|---|---|---|---|---|---|---|---|---|
| $L_3$ | 100 | 100 | 100 | 100 | 100 | 100 | 100 | 100 | 100 | 100 |
| $L_4$ | 22 | 100 | 100 | 100 | 100 | 100 | 100 | 100 | 100 | 100 |
| $L_5$ | 11 | 51 | 100 | 100 | 100 | 100 | 100 | 100 | 100 | 100 |
| $L_6$ | 10 | 8 | 37 | 100 | 98 | 100 | 100 | 100 | 100 | 100 |
| $L_7$ | 8 | 8 | 20 | 49 | 100 | 100 | 100 | 100 | 98 | 100 |
| $L_8$ | 5 | 6 | 19 | 52 | 75 | 95 | 100 | 100 | 100 | 100 |
| $L_9$ | 7 | 7 | 6 | 24 | 29 | 66 | 95 | 99 | 98 | 93 |
| $L_{10}$ | 2 | 2 | 2 | 6 | 12 | 49 | 77 | 96 | 96 | 98 |
| $L_{11}$ | 2 | 2 | 3 | 3 | 6 | 11 | 46 | 31 | 96 | 89 |
| $L_{12}$ | 0 | 0 | 0 | 1 | 2 | 4 | 21 | 30 | 27 | 62 |

Figure 4: Experimental results. Corollary 5.2 predicts that a transformer with depth $k$ can recognize language $L_{k+2}$ but not $L_{k+3}$ (demarcated by the black line). Up to at least $L_{12}$, this closely predicts our experimental results (shown as numbers and colors).

$L_k$ language serves as a minimal testbed for depth separation because it represents the simplest form of sequential dependency (ordering of symbols) using only an alphabet of size 2.

## 5.1 Problem

Our experimental setup differs slightly from the framework presented above. For the *language recognition problem* considered above, training a transformer would require data containing both positive and negative examples, with the distribution of negative examples potentially having an important impact on learnability. Following Bhattamishra et al. (2020) and Huang et al. (2025), we reframe $L_k$ as a *next-token prediction problem*: For each prefix of a string, output the set of possible next symbols of the string.

Our data consist of source–target pairs, where the source is a string in $L_k$, preceded by a beginning-of-string symbol <BOS>, and the target is a string of the same length. Each target symbol is a code standing for the set of possible next source symbols. If $k$ is odd, for example, $L_3 = a^+ b^+ a^+$, then after <BOS>, there is only one possible set, $\{a\}$ (coded as 0), and after subsequent symbols, there are two possible sets, $\{a, b\}$ (coded as 0) if the string is in $a^+ b^*$, and $\{a, \text{<EOS>}\}$ (coded as 1) if the string is in $a^+ b^+ a^+$ (where <EOS> stands for the end of the string). An example source–target pair is:

$$S = \text{<BOS>}\, a\, a\, a\, b\, b\, b\, b\, a\, a\, a\, a\, a$$
$$T = \quad 0 \quad\ 0\, 0\, 0\, 0\, 0\, 0\, 0\, 1\, 1\, 1\, 1\, 1$$

If $k$ is even, the possible sets would be $\{a\}$ (coded as 0) after <BOS>, and $\{a, b\}$ (coded as 0) and $\{\text{<EOS>}, b\}$ (coded as 1) subsequently. It turns out that for $L_k$, the output for a prefix <BOS> $\cdot\, w[1 : i]$ should be 1 if and only if $w[1 : i] \in L_k$.

We can define the next-token prediction problem for $\mathsf{TL}[\overleftarrow{\#}]$ in the same way, but without <BOS>:

**Definition 5.1** (Next-Token Prediction Problem for $L_k$). *We say a $\mathsf{TL}[\overleftarrow{\#}]$ formula $\phi$ can solve the next-token prediction problem for $L_k$ if for all $w \in L_k$ and $1 \le i \le |w|$, we have $w[1 : i] \models \phi \iff$*

$w[1:i] \in L_k$. *That is, $w[1:i] \models \phi$ if the prediction is 1, while $w[1:i] \not\models \phi$ if the prediction is 0.*
*Note that, unlike in recognition, we only consider prefixes of strings that are in $L_k$.*

The depth hierarchy from Theorem 4.10 can be adapted to the next-token prediction problem for $L_k$.

**Corollary 5.2** (Corollary of Theorem 4.10). *A depth-$(k+1)$ $\mathsf{TL}[\overleftarrow{\#}]$ formula can solve the next-token prediction problem for $L_{k+3}$, but no depth-$k$ $\mathsf{TL}[\overleftarrow{\#}]$ formula can.*

*Proof.* See Appendix C.4. □

### 5.2 Setup

We generated samples of $L_k$ to place into bins $[201, 250]$, $[251, 300]$, $[301, 350]$, $[351, 400]$ by uniformly sampling a length $n$ from the bin and uniformly sampling $k-1$ positions at which to switch between $a$ and $b$. For each $k$ and each bin, 1000 strings were generated. The $[201, 250]$ bin of 1000 examples was split into a training set of 800 examples and a validation set of 200 examples. The other bins were reserved for evaluation.

We trained future-masked transformers without positional encodings. Because the sets of next tokens are mutually exclusive, we trained the transformer to perform multi-class classification with cross-entropy as the loss function. Adam was used as the optimizer (Kingma and Ba, 2015). The dimension $d$ and learning rate $\eta$ were tuned by searching over $d \in [256, 512]$ and $\eta \in [10^{-4}, 10^{-5}]$. Each hyperparameter configuration was trained for 25 epochs or until 100% accuracy was achieved on the validation set. Then we evaluated the trained model on the test sets, considering the transformer to have made a correct prediction if and only if its prediction matched the target at every single position. The experiments were run on an internal cluster of GPUs. Performing the training loop for a given number of layers over all $L_k$ required an average of $9.37 \cdot 10^4$ TFLOPs and 936.8 MiB of memory.

### 5.3 Results

Figure 4 shows the final accuracies of models with varying depth on $L_k$ with varying $k$. Corollary 5.2 predicts that a $\mathsf{TL}[\overleftarrow{\#}]$ formula must have depth at least $k$ in order to solve the next-token prediction problem for $L_{k+2}$. In most cases, the transformer obtains 100% accuracy when Corollary 5.2 predicts it, and even generalizes to lengths up to double the training length. Other factors, like width, data diversity, and training dynamics of deeper transformers, may also play a role in practice.

## 6 Limitations

Our theoretical results apply to fixed-precision transformers with and without positional encodings, whose definition differs subtly from both standard real-valued softmax transformers and fixed-precision transformers considered in previous work. Our experimental results did not use positional encodings because we expect that extremely long input lengths are required to see our negative results apply. Additionally, our experiments only concern formal language tasks – namely, the languages $L_k$.

## 7 Conclusion

This paper adds to the growing list of exact equivalences between variants of transformers and logics or complexity classes (Yang et al., 2024; Merrill and Sabharwal, 2024; Li et al., 2024; Li and Cotterell, 2025). Here, we have shown that transformers that round to fixed precision except inside attention are exactly equivalent to $\mathsf{TL}[\overleftarrow{\#}]$ and C-RASP. Moreover, we have proven a strict depth hierarchy for $\mathsf{TL}[\overleftarrow{\#}]$, which implies a strict depth hierarchy for (this variant of) transformers. Unlike previous depth separations for softmax transformers (Sanford et al., 2024; Chen et al., 2025), our results apply to parameter-uniform transformers and so are particularly relevant to length generalization. Future work on the experimental side could look for real-world phenomena that involve sequential dependencies like those in $L_k$ and study how well language models handle them.

## Acknowledgements

This material is based in part upon work supported by the National Science Foundation under Grant No. 2502292 and a Graduate Research Fellowship under Grant No. 2236418. Any opinions, findings, and conclusions or recommendations expressed in this material are those of the authors and do not necessarily reflect the views of the National Science Foundation. We would like to thank Dana Angluin and Lena Strobl for their generous input on the presentation of the theoretical results and development of the experiments in this paper. We also thank Michael Hahn for introducing us to the fascinating connection between $\text{TL}[\overleftarrow{\#}, \overrightarrow{\#}]$ and $\widehat{\text{MAJ}}_2[<]$, which paved the way towards the results proven here, and Gavin Dooley, Peter Cholak, and Anand Pillay for insightful conversations about $\text{TL}[\overleftarrow{\#}]$. Finally, we thank the anonymous reviewers for their helpful comments.

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

# A  Logic Preliminaries

## A.1  Temporal Logics with Counting

**Definition A.1.** *The syntax of* $\mathsf{TL}[\overleftarrow{\#}]$ *is as follows:*

$$\phi ::= Q_\sigma \mid t_1 < t_2 \mid \neg\phi_1 \mid \phi_1 \wedge \phi_2 \qquad \sigma \in \Sigma \qquad \textit{Boolean-valued formulas}$$
$$t ::= \overleftarrow{\#}[\phi_1] \mid t_1 + t_2 \mid 1 \qquad\qquad\qquad\qquad \textit{integer-valued terms}$$

*The syntax of* $\mathsf{TL}[\overleftarrow{\#}, \overrightarrow{\#}]$ *additionally has counting terms* $t ::= \overrightarrow{\#}[\phi]$. *The semantics of formulas is defined as follows:*

$$w, i \models Q_\sigma \qquad \Longleftrightarrow \qquad w_i = \sigma \tag{3a}$$

$$w, i \models \neg\phi \qquad \Longleftrightarrow \qquad w, i \not\models \phi \tag{3b}$$

$$w, i \models \phi_1 \wedge \phi_2 \quad \Longleftrightarrow \quad w, i \models \phi_1 \text{ and } w, i \models \phi_2 \tag{3c}$$

$$w, i \models t_1 < t_2 \quad \Longleftrightarrow \quad t_1^{w,i} < t_2^{w,i}. \tag{3d}$$

*The semantics of terms is defined as follows:*

$$\overleftarrow{\#}[\phi]^{w,i} = |\{j \in [1, i] \mid w, j \models \phi\}| \tag{4a}$$

$$\overrightarrow{\#}[\phi]^{w,i} = |\{j \in [i, |w|] \mid w, j \models \phi\}| \tag{4b}$$

$$(t_1 + t_2)^{w,i} = t_1^{w,i} + t_2^{w,i} \tag{4c}$$

$$1^{w,i} = 1. \tag{4d}$$

*We write* $w \models \phi$ *if* $w, |w| \models \phi$, *and we say that* $\phi$ *defines the language* $\mathcal{L}(\phi) = \{w \mid w \models \phi\}$.

**Definition A.2.** *The* depth *of formulas and terms of* $\mathsf{TL}[\overleftarrow{\#}]$ *and* $\mathsf{TL}[\overleftarrow{\#}, \overrightarrow{\#}]$ *is defined by:*

$$\mathrm{dp}(Q_\sigma) = 0$$
$$\mathrm{dp}(\neg\phi) = \mathrm{dp}(\phi)$$
$$\mathrm{dp}(\phi_1 \wedge \phi_2) = \max\{\mathrm{dp}(\phi_1), \mathrm{dp}(\phi_2)\}$$
$$\mathrm{dp}(t_1 < t_2) = \max\{\mathrm{dp}(t_1), \mathrm{dp}(t_2)\}$$
$$\mathrm{dp}(\overleftarrow{\#}[\phi]) = \mathrm{dp}(\overrightarrow{\#}[\phi]) = \mathrm{dp}(\phi) + 1$$
$$\mathrm{dp}(t_1 + t_2) = \max\{\mathrm{dp}(t_1), \mathrm{dp}(t_2)\}$$
$$\mathrm{dp}(1) = 0.$$

*We write* $\mathsf{TL}[\overleftarrow{\#}]_k$ *(or* $\mathsf{TL}[\overleftarrow{\#}, \overrightarrow{\#}]_k$*) for the set of all* $\mathsf{TL}[\overleftarrow{\#}]$ *(or* $\mathsf{TL}[\overleftarrow{\#}, \overrightarrow{\#}]$, *resp.) formulas with depth at most* $k$.

We will often assume that any comparison $t_1 < t_2$ can be written in the form $\sum_{\chi \in \mathcal{L}} \lambda_\chi \overleftarrow{\#}[\chi] \geq C$ or $\sum_{\chi \in \mathcal{L}} \lambda_\chi \overleftarrow{\#}[\chi] + \sum_{\chi \in \mathcal{R}} \lambda_\chi \overrightarrow{\#}[\chi] \geq C$ where $\mathcal{L}, \mathcal{R}$ are sets of formulas and $\lambda_\ell, C \in \mathbb{Z}$.

## A.2  Examples of $\mathsf{TL}[\overleftarrow{\#}]$

Recall from Example 2.2 that the Dyck language is defined by the formula
$$\phi_{\mathrm{Dyck}} = (\overleftarrow{\#}[Q_(] = \overleftarrow{\#}[Q_)]) \wedge (\overleftarrow{\#}[\overleftarrow{\#}[Q_(] < \overleftarrow{\#}[Q_)]] = 0).$$

The table below shows how this formula works for the string $(())()$, which belongs to the Dyck language.

| subformula | description | ( | ( | ) | ) | ( | ) |
|---|---|---|---|---|---|---|---|
| $Q_($ | is left paren | $\top$ | $\top$ | $\bot$ | $\bot$ | $\top$ | $\bot$ |
| $Q_)$ | is right paren | $\bot$ | $\bot$ | $\top$ | $\top$ | $\bot$ | $\top$ |
| $\overleftarrow{\#}[Q_(]$ | num of left parens | 1 | 2 | 2 | 2 | 3 | 3 |
| $\overleftarrow{\#}[Q_)]$ | num of right parens | 0 | 0 | 1 | 2 | 2 | 3 |
| $\overleftarrow{\#}[Q_(] = \overleftarrow{\#}[Q_)]$ | balanced | $\bot$ | $\bot$ | $\bot$ | $\top$ | $\bot$ | $\top$ |
| $\overleftarrow{\#}[Q_(] < \overleftarrow{\#}[Q_)]$ | violates matching | $\bot$ | $\bot$ | $\bot$ | $\bot$ | $\bot$ | $\bot$ |
| $\overleftarrow{\#}[\overleftarrow{\#}[Q_(] < \overleftarrow{\#}[Q_)]]$ | num of violations | 0 | 0 | 0 | 0 | 0 | 0 |
| $\overleftarrow{\#}[\overleftarrow{\#}[Q_(] < \overleftarrow{\#}[Q_)]] = 0$ | matched | $\top$ | $\top$ | $\top$ | $\top$ | $\top$ | $\top$ |
| $\overleftarrow{\#}[\overleftarrow{\#}[Q_(] < \overleftarrow{\#}[Q_)]] = 0 \wedge \overleftarrow{\#}[Q_(] = \overleftarrow{\#}[Q_)]$ | matched and balanced | $\bot$ | $\bot$ | $\bot$ | $\top$ | $\bot$ | $\top$ |

The table below shows how this formula works for the string ())()(, which does not belong to the Dyck language.

| subformula | description | ( | ) | ) | ( | ) | ( |
|---|---|---|---|---|---|---|---|
| $Q_($ | is left paren | ⊤ | ⊥ | ⊥ | ⊤ | ⊥ | ⊤ |
| $Q_)$ | is right paren | ⊥ | ⊤ | ⊤ | ⊥ | ⊤ | ⊥ |
| $\overleftarrow{\#}[Q_(]$ | num of left parens | 1 | 1 | 1 | 2 | 2 | 3 |
| $\overleftarrow{\#}[Q_)]$ | num of right parens | 0 | 1 | 2 | 2 | 3 | 3 |
| $\overleftarrow{\#}[Q_(] = \overleftarrow{\#}[Q_)]$ | balanced | ⊥ | ⊤ | ⊥ | ⊤ | ⊥ | ⊤ |
| $\overleftarrow{\#}[Q_(] < \overleftarrow{\#}[Q_)]$ | violates matching | ⊥ | ⊥ | ⊤ | ⊥ | ⊤ | ⊥ |
| $\overleftarrow{\#}[\overleftarrow{\#}[Q_(] < \overleftarrow{\#}[Q_)]]$ | num of violations | 0 | 0 | 1 | 1 | 2 | 2 |
| $\overleftarrow{\#}[\overleftarrow{\#}[Q_(] < \overleftarrow{\#}[Q_)]] = 0$ | matched | ⊤ | ⊤ | ⊥ | ⊥ | ⊥ | ⊥ |
| $\overleftarrow{\#}[\overleftarrow{\#}[Q_(] < \overleftarrow{\#}[Q_)]] = 0 \land \overleftarrow{\#}[Q_(] = \overleftarrow{\#}[Q_)]$ | matched and balanced | ⊥ | ⊤ | ⊥ | ⊥ | ⊥ | ⊥ |

## A.3  Extensions to $\mathsf{TL}[\overleftarrow{\#}, \overrightarrow{\#}]$

We will often make use of the following operator in $\mathsf{TL}[\overleftarrow{\#}, \overrightarrow{\#}]$, which does not increase its expressive power or affect the depth of formulas, but saves space when writing.

$$(\phi \; ? \; t_{\text{then}} \; \mathbf{:} \; t_{\text{else}})^{w,i} = \begin{cases} t_{\text{then}} & w, i \models \phi \\ t_{\text{else}} & w, i \not\models \phi \end{cases}$$

**Lemma A.3** (Yang and Chiang 2024)**.** *Any formula $\phi$ of $\mathsf{TL}[\overleftarrow{\#}, \overrightarrow{\#}]$ that uses the ? operator can be converted into a formula that does not use the ? operator, defines the same language as $\phi$, and has the same depth as $\phi$.*

*Proof.* Any comparison formula involving the ? operator can be written in the form

$$(\psi_{\text{if}} \; ? \; t_{\text{then}} \; \mathbf{:} \; t_{\text{else}}) + \sum_{\ell \in [m]} t_\ell \geq C,$$

which can be rewritten as

$$\left( \psi_{\text{if}} \land t_{\text{then}} + \sum_{\ell \in [m]} t_\ell \geq C \right) \lor \left( \neg\psi_{\text{if}} \land t_{\text{else}} + \sum_{\ell \in [m]} t_\ell \geq C \right).$$

This rule can be used iteratively to rewrite all the ? operators out of a formula. □

It can also be convenient to allow an unmasked counting operator # and strict counting operators $\overset{\circ}{\overleftarrow{\#}}$ and $\overset{\circ}{\overrightarrow{\#}}$, which do not count the current position.

$$\#[\phi]^{w,i} = |\{j \in [1, |w|] \mid w, j \models \phi\}|$$
$$\overset{\circ}{\overleftarrow{\#}}[\phi]^{w,i} = |\{j \in [i, |w| - 1] \mid w, j \models \phi\}|$$
$$\overset{\circ}{\overrightarrow{\#}}[\phi]^{w,i} = |\{j \in [i + 1, |w|] \mid w, j \models \phi\}|$$

**Lemma A.4.** *Any formula $\phi$ of $\mathsf{TL}[\overleftarrow{\#}, \overrightarrow{\#}]$ that uses #, $\overset{\circ}{\overleftarrow{\#}}$ or $\overset{\circ}{\overrightarrow{\#}}$ can be converted into a formula that does not use #, $\overset{\circ}{\overleftarrow{\#}}$, or $\overset{\circ}{\overrightarrow{\#}}$, defines the same language as $\phi$, and has the same depth as $\phi$.*

*Proof.* These counting terms can be rewritten equivalently using $\overleftarrow{\#}$ and $\overrightarrow{\#}$:

$$\#[\phi] \equiv \overleftarrow{\#}[\phi] + \overrightarrow{\#}[\phi] - (\phi \; ? \; 1 \; \mathbf{:} \; 0)$$
$$\overset{\circ}{\overleftarrow{\#}}[\phi] \equiv \overleftarrow{\#}[\phi] - (\phi \; ? \; 1 \; \mathbf{:} \; 0)$$
$$\overset{\circ}{\overrightarrow{\#}}[\phi] \equiv \overrightarrow{\#}[\phi] - (\phi \; ? \; 1 \; \mathbf{:} \; 0).$$
□

## A.4  Proof of Lemma 2.9 (definability of piecewise testable languages)

**Lemma 2.9.** *Any $k$-piecewise testable language is definable in $\mathsf{TL}[\overleftarrow{\#}]_k$, and any $(2k + 1)$-piecewise testable language is definable in $\mathsf{TL}[\overleftarrow{\#}, \overrightarrow{\#}]_{k+1}$.*

Any $k$-piecewise testable language $L$ can, by definition, be written as a Boolean combination of $\mathcal{J}$-expressions of the form

$$\Sigma^* \sigma_1 \Sigma^* \sigma_2 \Sigma^* \cdots \Sigma^* \sigma_k \Sigma^*.$$

This is defined by the $\mathsf{TL}[\overleftarrow{\#}]$ formula of depth $k$:

$$\phi = \overleftarrow{\#}[\cdots \overleftarrow{\#}[\overleftarrow{\#}[Q_{\sigma_1} \geq 1] \wedge Q_{\sigma_2}] \geq 1 \cdots \wedge Q_{\sigma_k}] \geq 1.$$

Any $(2k+1)$-piecewise testable language $L$ can, by definition, be written as a Boolean combination of $\mathcal{J}$-expressions of the form

$$\Sigma^* \sigma_1 \Sigma^* \sigma_2 \Sigma^* \cdots \Sigma^* \sigma_{2k+1} \Sigma^*.$$

We need to show that this is expressible with $k+1$ nestings of $\overleftarrow{\#}$ and $\overrightarrow{\#}$. We do this by first finding the middle symbol $\sigma_{k+1}$ and checking to the left and right that the correct symbols appear. First, define depth-$k$ subformulas that check the left and right halves of the $\mathcal{J}$-expression:

$$\phi_L = \overleftarrow{\#}[Q_{\sigma_k} \wedge \overleftarrow{\#}[Q_{\sigma_{k-1}} \wedge \overleftarrow{\#}[\cdots \overleftarrow{\#}[Q_{\sigma_1}] \geq 1 \cdots] \geq 1] \geq 1] \geq 1$$
$$\phi_R = \overrightarrow{\#}[Q_{\sigma_{k+2}} \wedge \overrightarrow{\#}[Q_{\sigma_{k+3}} \wedge \overrightarrow{\#}[\cdots \overrightarrow{\#}[Q_{\sigma_{2k+1}}] \geq 1 \cdots] \geq 1] \geq 1] \geq 1.$$

Then, $L$ is defined by the $\mathsf{TL}[\overleftarrow{\#}, \overrightarrow{\#}]$ formula of depth $(k+1)$:

$$\phi = \overleftarrow{\#}[\phi_L \wedge Q_{\sigma_{k+1}} \wedge \phi_R] \geq 1.$$

# B  Transformer Equivalence

In this section and below, the expression $\mathbb{I}[\cdot]$ has the value 1 if the statement inside the brackets is true, and 0 otherwise.

## B.1  Transformers

**Definition B.1.** *A* fixed-precision number *with $p$ total bits and $s$ fractional bits is a rational number of the form $m \cdot 2^{-s}$ where $m$ is an integer and $-2^{p-1} \leq m < 2^{p-1}$. We write $\mathbb{F}_{p,s}$, or simply $\mathbb{F}$, for the set of all fixed-precision numbers with $p$ total bits and $s$ fractional bits.*

*We represent negative numbers using two's complement. If $b \in [p]$, the $b$-th bit of a fixed-precision number $x$, written $\langle x \rangle_b$, is defined as*

$$\langle x \rangle_b = \begin{cases} 1 & \text{if } \lfloor x/2^{b-s-1} \rfloor \text{ is odd} \\ 0 & \text{otherwise.} \end{cases}$$

*If $x$ is a real number, we write $\mathrm{round}_{\mathbb{F}}(x)$ or simply $\mathrm{round}(x)$ for the greatest element of $\mathbb{F}$ less than or equal to $x$.*

The following definition abstracts away from a number of details, but suffices for our purposes (which is to prove Proposition B.6).

**Definition B.2.** *A* future-masked fixed-precision transformer *of depth $k$ is a function $T : \Sigma^* \to \mathbb{F}$, defined in terms of functions*

$$E : \Sigma \to \mathbb{F}^d$$
$$W_Q^{(\ell)}, W_K^{(\ell)}, W_V^{(\ell)}, f^{(\ell)} : \mathbb{F}^d \to \mathbb{F}^d \qquad \ell = 1, \dots, k$$
$$W_{\text{out}} : \mathbb{F}^d \to \mathbb{F}.$$

*On input $w$, $T(w)$ is computed as follows:*

$$\mathbf{h}_i^{(0)}(w) = E(w_i) \tag{5}$$

*For $\ell = 1, \ldots, k$:*

$$\mathbf{q}_i^{(\ell)}(w) = W_Q^{(\ell)}\left(\mathbf{h}_i^{(\ell-1)}(w)\right) \tag{6}$$

$$\mathbf{k}_i^{(\ell)}(w) = W_K^{(\ell)}\left(\mathbf{h}_i^{(\ell-1)}(w)\right) \tag{7}$$

$$\mathbf{v}_i^{(\ell)}(w) = W_V^{(\ell)}\left(\mathbf{h}_i^{(\ell-1)}(w)\right) \tag{8}$$

$$\mathbf{s}_{ij}^{(\ell)}(w) = \mathbf{q}_i^{(\ell)}(w) \cdot \mathbf{k}_j^{(\ell)}(w) \tag{9}$$

$$\mathbf{c}_i^{(\ell)}(w) = \mathrm{round}\left(\frac{\sum_{j=1}^i \mathrm{round}\left(\exp\left(\mathbf{s}_{ij}^{(\ell)}(w)\right)\mathbf{v}_j^{(\ell)}(w)\right)}{\sum_{j=1}^i \mathrm{round}\left(\exp\left(\mathbf{s}_{ij}^{(\ell)}(w)\right)\right)}\right) \tag{10}$$

$$\mathbf{h}_i^{(\ell)}(w) = f^{(\ell)}\left(\mathbf{c}_i^{(\ell)}(w) + \mathbf{h}_i^{(\ell-1)}(w)\right) \tag{11}$$

*where we say Eq. (10) evaluates to the average of all $\mathbf{v}_j^{(\ell)}$ if the denominator is $0$, and finally*

$$T(w) = W_{\mathrm{out}}\left(\mathbf{h}_{|w|}^{(k)}(w)\right). \tag{12}$$

*We say that $T$ accepts $w$ if $T(w) > 0$.*

Note crucially that Eq. (10) is written so that even if $i \gg 2^s$, it is still possible to obtain nonzero values.

## B.2 Proof of Theorem 3.1

**Theorem 3.1.** *A language $L$ is defined by a formula of $\mathsf{TL}[\overset{\leftarrow}{\#}]$ of depth $k$ if and only if $\texttt{<BOS>} \cdot L$ is recognized by a fixed-precision transformer of depth $k$.*

We first define what it means for a fixed-precision transformer and a formula to simulate each other.

**Definition B.3.** *We say that a $\mathsf{TL}[\overset{\leftarrow}{\#}]$ formula $\phi$ simulates a fixed-precision transformer $T$ if, for all $w \in \Sigma^*$,*

$$w, i \models \phi \iff T(\texttt{<BOS>} \cdot w)_i > 0.$$

*In other words, $w \models \phi$ if and only if $T$ accepts $\texttt{<BOS>} \cdot w$.*

*We say that a fixed-precision transformer $T$ with depth $k$ and dimension $d$ simulates a formula $\phi$ of $\mathsf{TL}[\overset{\leftarrow}{\#}]$ $T$ if, for all $w \in \Sigma^*$,*

$$T(\texttt{<BOS>} \cdot w)_i > 0 \iff w, i \models \phi.$$

*Again, $T$ accepts $\texttt{<BOS>} \cdot w$ if and only if $w \models \phi$.*

We prove the two directions of Theorem 3.1 separately: from $\mathsf{TL}[\overset{\leftarrow}{\#}]$ to fixed-precision transformers in Proposition B.4, and from fixed-precision transformers to $\mathsf{TL}[\overset{\leftarrow}{\#}]$ in Proposition B.6.

**Proposition B.4.** *Let $\phi$ be a $\mathsf{TL}[\overset{\leftarrow}{\#}]$ formula of depth $k$. There exists a fixed-precision transformer $T_\phi$ of depth $k$ which simulates $\phi$.*

*Proof.* This was essentially shown by Yang and Chiang (2024), but they simulated $\mathsf{TL}[\overset{\leftarrow}{\#}]$ using infinite-precision transformers with layer normalization. Here, we modify the proof to use rounding instead of layer normalization to simulate comparison operations.[3]

The case that differs is that of a subformula $\psi = \sum_{\chi \in \mathcal{L}} \lambda_\chi \overset{\leftarrow}{\#}[\chi] \geq C$. Assume that previous layers have computed $\mathbb{I}[w, j \models \chi]$ for $\chi \in \mathcal{L}$ at all positions $j \in [n]$. We want to construct a new layer that computes $\mathbb{I}[w, i \models \psi]$ at all positions $i \in [n]$. Use uniform attention and construct the value projection $W_V$ so that

$$\mathbf{v}_j = \sum_{\chi \in \mathcal{L}} \lambda_\chi \mathbb{I}[w, j \models \chi] - C\,\mathbb{I}[w_j = \texttt{<BOS>}].$$

---

[3]A further difference is that we store Boolean values as 0 for false and 1 for true, while Yang and Chiang (2024) used $\left[\begin{smallmatrix} +1 \\ -1 \end{smallmatrix}\right]$ for false and $\left[\begin{smallmatrix} -1 \\ +1 \end{smallmatrix}\right]$ for true. They used this more complicated encoding in order to deal with layer normalization, which is unnecessary here. Our proof could be modified straightforwardly to accommodate layernorm by using this representation.

After averaging, we get

$$\mathbf{c}_i = \frac{1}{i+1}\left(\sum_{\chi \in \mathcal{L}}\left(\lambda_\chi \sum_{j=1}^{i}\mathbb{I}[w, j \models \chi]\right) - C\right)$$

which rounds to $-2^{-s}$ or below if $\phi$ is false, and rounds to 0 or above if $\phi$ is true. We can then use the FFNN $f$ to map these two cases to 0 or 1, respectively.

If $\phi$ is the entire formula, the output function $W_{\text{out}}$ just takes the computed value $\mathbb{I}[w, j \models \phi]$ and maps 0 and 1 to $-1$ and $+1$, respectively. $\qquad\square$

The following lemma will be used repeatedly.

**Lemma B.5** (Chiang et al., 2023). *If $F\colon \Sigma^* \to \mathbb{F}^*$ is length-preserving, $g\colon \mathbb{F} \to \mathbb{F}$, and there are formulas $\phi_{\langle F \rangle_b}$ such that*

$$w, i \models \phi_{\langle F \rangle_b} \iff \langle F(w)_i \rangle_b = 1$$

*then there is a formula $\phi_{\langle g(F) \rangle_b}$ such that*

$$w, i \models \phi_{\langle g(F) \rangle_b} \iff \langle g(F(w)_i) \rangle_b = 1.$$

*Similarly if $g$ is a function of more than one fixed-precision number.*

**Proposition B.6.** *Let $T$ be a fixed-precision transformer of depth $k$. There exists a $\mathsf{TL}[\overleftarrow{\#}]$ formula $\phi_T$ of depth $k$ which simulates $T$.*

*Proof.* We will show that for every activation $\mathbf{h}_{i,c}^{(\ell)}$ and $b \in [p]$, there is some formula $\phi_{\langle \mathbf{h}_c^{(\ell)} \rangle_b}$ such that

$$w, i \models \phi_{\langle \mathbf{h}_c^{(\ell)} \rangle_b} \iff \langle \mathbf{h}_{i,c}^{(\ell)}(w) \rangle_b = 1. \tag{13}$$

The construction proceeds by induction on $\ell$. For $\ell = 0$, define the word embedding as

$$\phi_{\langle \mathbf{h}_c^{(0)} \rangle_b} = \bigvee_{\substack{\sigma \in \Sigma \\ \langle E(\sigma)_c \rangle_b = 1}} Q_\sigma$$

so that Eq. (13) holds for $\ell = 0$.

Now suppose that Eq. (13) holds for layer $\ell$, and consider layer $(\ell + 1)$. Use Lemma B.5 on $W^{\mathrm{Q}}$, $W_{\mathrm{K}}$, and $W_{\mathrm{V}}$ to obtain formulas $\phi_{\langle \mathbf{q}_c^{(\ell)} \rangle_b}$, $\phi_{\langle \mathbf{k}_c^{(\ell)} \rangle_b}$, and $\phi_{\langle \mathbf{v}_c^{(\ell)} \rangle_b}$ such that

$$w, i \models \phi_{\langle \mathbf{q}_c^{(\ell)} \rangle_b} \iff \langle \mathbf{q}_{i,c}^{(\ell)}(w) \rangle_b = 1 \tag{14}$$

$$w, i \models \phi_{\langle \mathbf{k}_c^{(\ell)} \rangle_b} \iff \langle \mathbf{k}_{i,c}^{(\ell)}(w) \rangle_b = 1 \tag{15}$$

$$w, i \models \phi_{\langle \mathbf{v}_c^{(\ell)} \rangle_b} \iff \langle \mathbf{v}_{i,c}^{(\ell)}(w) \rangle_b = 1. \tag{16}$$

Equation (14) in particular allows us to write formulas $\phi_{\mathbf{q}^{(\ell)} = \vec{q}}$ for each $\vec{q} \in \mathbb{F}^d$ such that

$$w, i \models \phi_{\mathbf{q}^{(\ell)} = \vec{q}} \iff \mathbf{q}_i^{(\ell)} = \vec{q}.$$

Next, we want to compute the summands in the numerator and denominator of Eq. (10). These depend on two positions ($i$ and $j$), whereas a formula of $\mathsf{TL}[\overleftarrow{\#}]$ only depends on one position. But since $\mathbf{q}_i$ can only take on finitely many values, we can enumerate all of its possible values. That is, use Lemma B.5 again to obtain formulas

$$w, j \models \alpha_{\vec{q},c,b}^{(\ell)} \iff \left\langle \mathrm{round}\left(\exp\left(\vec{q} \cdot \mathbf{k}_j^{(\ell)}(w)\right)\mathbf{v}_{j,c}^{(\ell)}(w)\right)\right\rangle_b = 1$$

$$w, j \models \beta_{\vec{q},b}^{(\ell)} \iff \left\langle \mathrm{round}\left(\exp\left(\vec{q} \cdot \mathbf{k}_j^{(\ell)}(w)\right)\right)\right\rangle_b = 1.$$

and then write counting terms $A_c^{(\ell)}$ and $B^{(\ell)}$ that represent the numerator and denominator of $\mathbf{c}_c^{(\ell)}$:

$$A_c^{(\ell)} = \sum_{\vec{q}}\left(\phi_{\mathbf{q}^{(\ell)} = \vec{q}} \; ? \; \left(-2^{p-1} \cdot \overleftarrow{\#}\left[\alpha_{\vec{q},c,p}^{(\ell)}\right] + \sum_{b \in [p-1]} 2^{b-1} \cdot \overleftarrow{\#}\left[\alpha_{\vec{q},c,b}^{(\ell)}\right]\right) : 0\right)$$

$$B^{(\ell)} = \sum_{\vec{q}}\left(\phi_{\mathbf{q}^{(\ell)} = \vec{q}} \; ? \; \left(-2^{p-1} \cdot \overleftarrow{\#}\left[\beta_{\vec{q},p}^{(\ell)}\right] + \sum_{b \in [p-1]} 2^{b-1} \cdot \overleftarrow{\#}\left[\beta_{\vec{q},b}^{(\ell)}\right]\right) : 0\right)$$

so that

$$(A_c^{(\ell)})^{w,i} = 2^s \sum_{j \leq i} \text{round}\left(\exp\left(\mathbf{q}_i^{(\ell)}(w) \cdot \mathbf{k}_j^{(\ell)}(w)\right) \mathbf{v}_{j,c}^{(\ell)}(w)\right)$$

$$(B^{(\ell)})^{w,i} = 2^s \sum_{j \leq i} \text{round}\left(\exp\left(\mathbf{q}_i^{(\ell)}(w) \cdot \mathbf{k}_j^{(\ell)}(w)\right)\right).$$

Next, to define $\mathbf{c}^{(\ell)}$, we need to divide the numerator $A_c^{(\ell)}$ by the denominator $B^{(\ell)}$. We note that $B^{(\ell)}$ will be nonnegative because exp always returns a nonnegative value. We assume $B^{(\ell)} > 0$, but the other case is similar. The sign bit is then defined by

$$\phi_{\langle \mathbf{c}_c^{(\ell)} \rangle_p} = A_c^{(\ell)} < 0.$$

The remaining bits can be defined, from most to least significant, using the grade-school algorithm for long division:

$$t_{p-1} = 2^s A_c^{(\ell)} + (\phi_{\langle \mathbf{c}_c^{(\ell)} \rangle_p} \; ? \; 2^{p-1} B^{(\ell)} : 0) \qquad \phi_{\langle \mathbf{c}_c^{(\ell)} \rangle_{p-1}} = t_{p-1} \geq 2^{p-2} B^{(\ell)}$$

$$t_{p-2} = t_{p-1} - (\phi_{\langle \mathbf{c}_c^{(\ell)} \rangle_{p-1}} \; ? \; 2^{p-2} B^{(\ell)} : 0) \qquad \phi_{\langle \mathbf{c}_c^{(\ell)} \rangle_{p-2}} = t_{p-2} \geq 2^{p-3} B^{(\ell)}$$

$$t_{p-3} = t_{p-2} - (\phi_{\langle \mathbf{c}_c^{(\ell)} \rangle_{p-2}} \; ? \; 2^{p-3} B^{(\ell)} : 0) \qquad \phi_{\langle \mathbf{c}_c^{(\ell)} \rangle_{p-3}} = t_{p-3} \geq 2^{p-4} B^{(\ell)}$$

$$\vdots$$

$$t_1 = t_2 - (\phi_{\langle \mathbf{c}_c^{(\ell)} \rangle_2} \; ? \; 2^1 B^{(\ell)} : 0) \qquad \phi_{\langle \mathbf{c}_c^{(\ell)} \rangle_1} = t_1 \geq 2^0 B^{(\ell)}$$

Finally, we use Lemma B.5 on $f$ to obtain formulas $\phi_{\langle \mathbf{h}_c^{(\ell)} \rangle_b}$ satisfying Eq. (13). $\qquad\square$

## C   Depth Hierarchy for $\mathsf{TL}[\overset{\leftarrow}{\#}]$

### C.1   Proof of Lemma 4.6 (Commutativity of depth 1)

**Lemma 4.6** (Commutativity of depth 1). *For any depth-1 formula $\phi$ of $\mathsf{TL}[\overset{\leftarrow}{\#}, \mathsf{PNP}]_1$ or $\mathsf{TL}[\overset{\leftarrow}{\#}, \overset{\rightarrow}{\#}, \mathsf{PNP}]_1$ and any affix restriction $(\lambda, \varrho)$ with $|\varrho(\vec{n})| \geq 1$ for all $\vec{n}$, if the PNPs of $\phi$ are constant on the middle of $(\lambda, \varrho)$, then $\mathcal{L}(\phi)$ is commutative on the middle of $(\lambda, \varrho)$.*

Let $n = |w|$ and $w' \in {}_\lambda \Sigma_\varrho^*$ with $\Psi(w) = \Psi(w')$. As such, there is a permutation $\pi \colon [n] \to [n]$ such that $w_i' = w_{\pi(i)}$ for all $i$, and if $i$ is a position in the prefix $(\lambda(\vec{n}))$ or suffix $(\varrho(\vec{n}))$, then $\pi(i) = i$. We want to show for depth-0 formulas $\chi$ that $w, \pi(i) \models \chi \iff w', i \models \chi$.

If $\chi = Q_\sigma$: Since $w_i' = w_{\pi(i)}$ we have $w, \pi(i) \models \chi \iff w', i \models \chi$.

If $\chi = \Pi$ for some Parikh numerical predicate $\Pi$: If $i$ is a position in the prefix or suffix, then $\pi(i) = i$, while if $i$ is a position in the middle, $\Pi$ is constant. In either case, $w, \pi(i) \models \chi \iff w', i \models \chi$.

If $\chi = \neg \chi_1$ or $\chi = \chi_1 \wedge \chi_2$ where $\chi_1, \chi_2$ have depth 0, then $w, \pi(i) \models \chi \iff w', i \models \chi$ follows from the semantics of $\neg$ and $\wedge$.

Any minimal depth-1 formula $\phi$ can be written, for finite sets of depth-0 formulas $\mathcal{L}$ and $\mathcal{R}$, as

$$\phi = \sum_{\chi \in \mathcal{L}} \lambda_\chi \overset{\leftarrow}{\#}[\chi] + \sum_{\chi \in \mathcal{R}} \lambda_\chi \overset{\rightarrow}{\#}[\chi] \geq C.$$

We showed above that for each $\chi$, we have $w, \pi(i) \models \chi \iff w', i \models \chi$. The $\overset{\leftarrow}{\#}$ terms count all positions, so $\overset{\leftarrow}{\#}[\chi]^{w,n} = \overset{\leftarrow}{\#}[\chi]^{w',n}$. The $\overset{\rightarrow}{\#}$ terms only count the last position, and $\pi(n) = n$, so $\overset{\rightarrow}{\#}[\chi]^{w,n} = \overset{\rightarrow}{\#}[\chi]^{w',n}$ for all $\chi$. Thus, $w \models \phi \iff w' \models \phi$.

Finally, if $\phi = \neg \phi_1$ or $\phi = \phi_1 \wedge \phi_2$ where $\phi_1, \phi_2$ have depth at most 1, then $w, \pi(i) \models \phi \iff w', i \models \phi$ again follows from the semantics of $\neg$ and $\wedge$.

## C.2 Proof of Lemma 4.8 (Cropping Lemma for $\mathsf{TL}[\overleftarrow{\#}]$)

**Lemma 4.8** (Cropping Lemma for $\mathsf{TL}[\overleftarrow{\#}]$). *For any formula $\phi$ of $\mathsf{TL}[\overleftarrow{\#},\mathsf{PNP}]$ and any accommodating family of intervals $I\colon \mathbb{N}^{|\Sigma|} \to \mathcal{I}(\mathbb{N}^{|\Sigma|})$ such that the PNPs of $\phi$ are constant on $I$, there exists an accommodating family of intervals $I'\colon \mathbb{N}^{|\Sigma|} \to \mathcal{I}(\mathbb{N}^{|\Sigma|})$ such that $I'(\vec{n})$ sticks only to the top (and no other side) of $I(\vec{n})$ for all $\vec{n} \in N^{|\Sigma|}$, and all of the minimal depth-1 subformulas (and PNPs) of $\phi$ are constant on $I'$. Additionally, there exists such an $I'$ such that $I'(\vec{n})$ sticks only to the right of $I(\vec{n})$.*

Let $\psi_1, \psi_2, \ldots, \psi_c$ be the minimal depth-1 subformulas of $\phi$. That is, for $\ell \in [c]$,

$$\psi_\ell = \left( \sum_{\chi \in \mathcal{L}_\ell} \lambda_\chi \overleftarrow{\#}[\chi] \right) \ge C_\ell.$$

Because any PNPs in each $\psi_\ell$ are constant on $I$, each $\psi_\ell$ defines a half-plane in $\overleftarrow{\#}[Q_a]$ and $\overleftarrow{\#}[Q_b]$, where $w, i \models \psi_\ell$ iff $w[1:i]$ lands in the corresponding half-plane. Then $\phi$ is a Boolean combination of these half-planes. Thus if $\Psi(w[1:i])$ and $\Psi(w'[1:i'])$ both land in the same half-planes, they will both satisfy $\phi$ or both not satisfy $\phi$.

We will show that for a desired size $\vec{s}$, there is $I(\vec{n})$ sufficiently large such that, there is a subinterval $I'(\vec{n})$ with size at least $\vec{s}$ sticking only to the top of $I(\vec{n})$ (and no other side).

Let $m$ be the minimum absolute slope of any non-horizontal boundary line, and let $h$ be the maximum $b$-intercept of any horizontal boundary line.

For any $\vec{s} = (s_a, s_b)$, let $\vec{s}' = (s'_a, s'_b) = (s_a + 2, s_b + 1)$, and choose $\vec{n}$ such that $I(\vec{n})$ has size at least $[3^c \max(s'_b/m, s'_a), h + s'_b]$. Our goal is to find a subinterval $I'(\vec{n})$ that has size $\vec{s}'$, sticks to the top of $I(\vec{n})$, and does not cross any boundary lines.

Let $I_c$ be an arbitrary subinterval of size $[3^c \max(s'_b/m, s'_a), s'_b]$ that sticks to the top of $I(\vec{n})$. We will prove by induction on $c$: Given an interval $I_c$ with size $[3^c \max(s'_b/m, s'_a), s'_b)]$ that sticks to the top of $I(\vec{n})$ and a set of $c$ boundary lines, there is a subinterval $I'(\vec{n})$ of size $\vec{s}'$ that sticks to the top of $I(\vec{n})$ and does not cross any boundary lines.

The base case $c = 0$ is trivial: There are no boundary lines to cross, and $s'_a \le \max(s'_b/m, s'_a)$, so choose any subinterval of $I_0$ with size $\vec{s}'$, and shrink it 1 unit from the left, bottom, and right to obtain a subinterval $I'(\vec{n})$ of size $\vec{s}$ that does not to stick to the left, right, or bottom of $I_0$.

If $c > 0$, take an arbitrary boundary line and call it $\ell$ and let $I_c$ be an interval with size $[3^c \max(s'_b/m, s'_a), s'_b)]$.

If $\ell$ is horizontal, it must have $b$-intercept at most $h$, so there must be at least $s'_b$ space above it inside $I(\vec{n})$. So $I_c$ does not cross $\ell$, and neither does any subinterval of $I_c$. Arbitrarily choose $I_{c-1}$ to be the middle third of $I_c$, and use the induction hypothesis on $I_{c-1}$ and the remaining boundary lines.

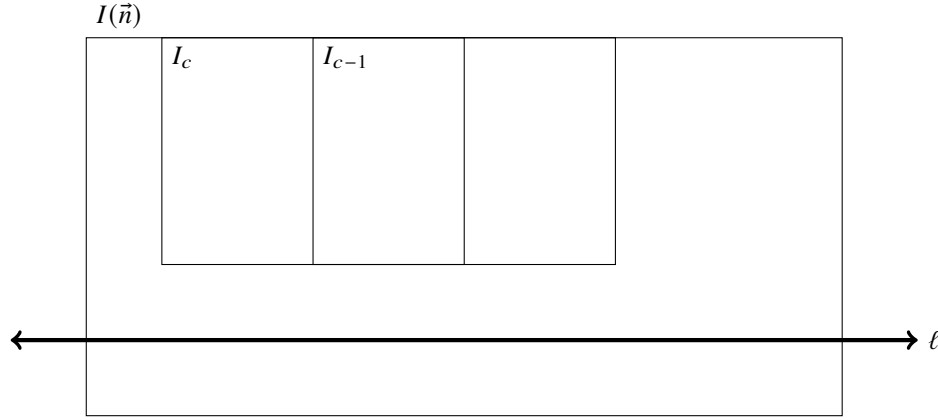

If $\ell$ is not horizontal, it must have absolute slope at least $m$. The part of $I_c$ that is crossed by $\ell$ must have width at most $s'_b/m$, whereas $I_c$ has width $3^c \max(s'_b/m, s'_a) \ge 3s'_b/m$, so either the left third

or right third of $I_c$ does not cross $\ell$. Choose $I_{c-1}$ to be that third, and use the induction hypothesis on $I_{c-1}$ and the remaining boundary lines.

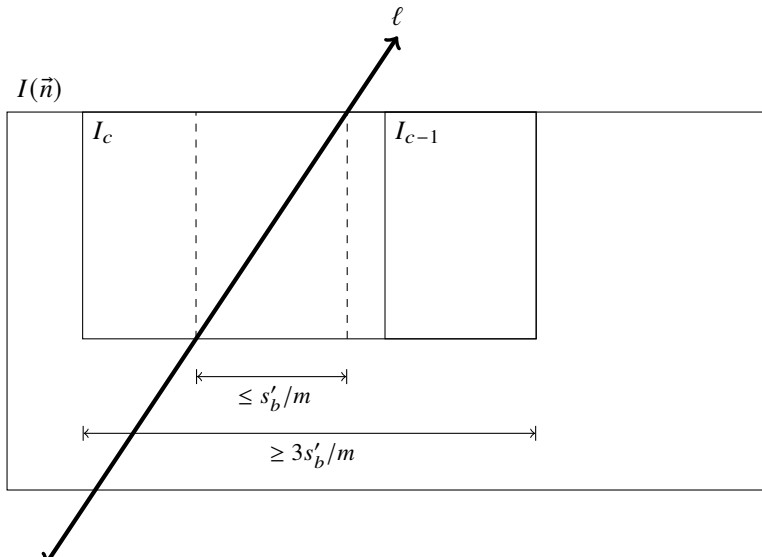

A similar argument (with $a$ and $b$ swapped) can be used to find an $I'(\vec{n})$ sticking only to the right of $I(\vec{n})$.

## C.3 Proof of Lemma 4.9 (Reduction Lemma)

**Lemma 4.9** (Reduction Lemma)**.** *For any depth-$k$ formula $\phi$ of* $\mathsf{TL}[\overleftarrow{\#}, \mathsf{PNP}]_k$ *(or* $\mathsf{TL}[\overleftarrow{\#}, \overrightarrow{\#}, \mathsf{PNP}]_k$*)* *and affix restriction $(\lambda, \varrho)$, if the PNPs and minimal depth-1 subformulas of $\phi$ are constant on the middle of $(\lambda, \varrho)$, then there is a formula $\phi'$ of depth $(k-1)$ of* $\mathsf{TL}[\overleftarrow{\#}, \mathsf{PNP}]_{k-1}$ *(or* $\mathsf{TL}[\overleftarrow{\#}, \overrightarrow{\#}, \mathsf{PNP}]_k$*,* *resp.) that defines $\lambda \mathcal{L}(\phi)_\varrho$, and the PNPs of $\phi'$ are constant on the middle of $(\lambda, \varrho)$.*

Let $\psi_1, \ldots, \psi_c$ be the minimal depth-1 subformulas of $\phi$. Each $\psi_\ell$ for $1 \le \ell \le c$ is of the form

$$\psi_\ell = \sum_{\chi \in \mathcal{L}_\ell} \lambda_\chi \overleftarrow{\#}[\chi] \ge C_\ell$$

if $\phi$ is a formula of $\mathsf{TL}[\overleftarrow{\#}, \mathsf{PNP}]_k$ or

$$\psi_\ell = \sum_{\chi \in \mathcal{L}_\ell} \lambda_\chi \overleftarrow{\#}[\chi] + \sum_{\chi \in \mathcal{R}_\ell} \lambda_\chi \overleftarrow{\#}[\chi] \ge C_\ell.$$

if $\phi$ is a formula of $\mathsf{TL}[\overleftarrow{\#}, \overrightarrow{\#}, \mathsf{PNP}]_k$.

In either case, $\psi_\ell$ is constant on the middle of $(\lambda, \varrho)$, meaning that for a given $\vec{n}$, each $\psi_\ell$ has the same truth value for $w \in {}_\lambda\Sigma^*_\varrho$ with $\Psi(w) = \vec{n}$ at positions $i$ such that $|\lambda(\vec{n})| \le i \le \|\vec{n}\| - |\rho(\vec{n})|$. The truth value of $\psi_\ell$, given the restriction to $(\lambda, \varrho)$, is determined solely by the Parikh vector of the word and the position $i$ at which it is evaluated. Thus there is a PNP $M_\ell$ which "hard-codes" the behavior of $\psi_\ell$. Moreover, $M_\ell$ is constant on the middle of $(\lambda, \varrho)$ because $\psi_\ell$ is. Thus we replace the depth-1 formula $\psi_\ell$ with the depth-0 formula $M_\ell$. Call the result $\phi_{\mathrm{red}}$.

The last step is to write a formula that checks if $w \in {}_\lambda\Sigma^*_\varrho$. For $\sigma \in \Sigma$, define a Parikh numerical predicate $\Pi_\sigma$ that is true at position $i$ if position $i$ belongs to the prefix/suffix and the symbol at that position of the prefix/suffix is $\sigma$:

$$w, i \models \Pi_\sigma \iff \begin{cases} \lambda(\Psi(w))_i = \sigma & i \le |\lambda(\Psi(w))| \\ \varrho(\Psi(w))_{i-(|w|-|\varrho(\Psi(w))|)} = \sigma & i \ge |w| - |\varrho(\Psi(w))| \\ \top & \text{otherwise.} \end{cases}$$

This is a Parikh numerical predicate because it is conditioned only on the position $i$ and the Parikh vector of $w$. Moreover, it is constant on the middle of $(\lambda, \varrho)$.

Then we can write the following formula, which checks whether $w \in {}_\lambda\Sigma_\varrho^*$ by checking whether $w$ at every position $i$ has the correct prefix/suffix:

$$\phi_{\text{aff}} = \left(\overleftarrow{\#}\left[(\Pi_a \wedge Q_a) \vee (\Pi_b \wedge Q_b)\right] = \overleftarrow{\#}[\top]\right)$$

Finally, define the new formula

$$\phi' = \phi_{\text{red}} \wedge \phi_{\text{aff}}$$

which has depth $(k-1)$ and defines ${}_\lambda L_\varrho$. Note that since $\phi_{\text{aff}}$ does not contain $\overrightarrow{\#}$, if $\phi$ did not contain $\overrightarrow{\#}$ then $\phi'$ does not either, so if $\phi \in \mathsf{TL}[\overleftarrow{\#}, \mathsf{PNP}]_k$ then $\phi' \in \mathsf{TL}[\overleftarrow{\#}, \mathsf{PNP}]_{k-1}$.

## C.4 Proof of Corollary 5.2

**Corollary 5.2** (Corollary of Theorem 4.10). *A depth-$(k+1)$ $\mathsf{TL}[\overleftarrow{\#}]$ formula can solve the next-token prediction problem for $L_{k+3}$, but no depth-$k$ $\mathsf{TL}[\overleftarrow{\#}]$ formula can.*

First, we show that the prediction problem for $L_{k+3}$ is solvable in $\mathsf{TL}[\overleftarrow{\#}]_{k+1}$. To decide whether to predict 0 or 1, we need to check that $w$ starts with $a$ and has exactly $(k+3)$ blocks so far: if so, predict 1; if not, predict 0. Since we may assume that the entire string $w$ belongs to $L_{k+3}$, we know that it starts with a block of $a$'s, and there are no more than $(k+2)$ blocks after that. We can check blocks 2 through $(k+2)$ by checking if $w$ belongs to $B_{k+1}$ (see Eq. (2)), and we can check the last block by testing whether the current symbol is $a$ (if $k$ is even) or $b$ (if $k$ is odd).

More formally, let $\phi_{B_{k+1}}$ define $B_{k+1}$ by Lemma 2.9, and let

$$\phi = \begin{cases} \phi_{B_{k+1}} \wedge Q_a & k \text{ is even} \\ \phi_{B_{k+1}} \wedge Q_b & k \text{ is odd.} \end{cases}$$

Then, for all $w \in L_{k+3}$ and $1 \le i \le |w|$ we have that

$$w[1:i] \in L_{k+3} \iff w[1:i] \models \phi.$$

In the other direction (the prediction problem for $L_{k+3}$ is not solvable in $\mathsf{TL}[\overleftarrow{\#}]_k$), suppose we had a depth-$k$ formula $\phi \in \mathsf{TL}[\overleftarrow{\#}]_k$ such that for all $w \in L_{k+3}$ and $1 \le i \le |w|$, $w[1:i] \models \phi \iff w[1:i] \in L_{k+3}$. Assume that $k$ is even (the odd case is similar). We can use Lemmas 4.8 and 4.9 just as in the proof of Theorem 4.10 to obtain an accommodating affix restriction $\lambda_1(\vec{n}) \in L_k$ and $\varrho_1(\vec{n}) \in a^+$, as well as a depth-1 formula $\phi_1 \in \mathsf{TL}[\overleftarrow{\#}, \mathsf{PNP}]_1$, which defines ${}_{\lambda_1}(L_{k+3})_{\varrho_1}$ and only uses PNPs which are constant over $(\lambda_1, \varrho_1)$.

Since $(\lambda_1, \varrho_1)$ is accommodating, choose $\vec{n}$ so that the middle has $s_a \ge 2$ occurrences of $a$ and $s_b \ge 2$ occurrences of $b$. Construct strings

$$w = \lambda_1(\vec{n}) b^{s_b-1} a^{s_a-1} b a \varrho_1(\vec{n})$$
$$w' = \lambda_1(\vec{n}) b^{s_b} a^{s_a} \varrho_1(\vec{n}) b a$$

which both belong to $L_{k+3}$ (because the prefix $\lambda_1(\vec{n})$ has $k$ blocks ending with a block of $b$'s and the suffix $\varrho_1(\vec{n})$ is all $a$'s, so both strings have $(k+3)$ blocks). Let $i = |\lambda_1(\vec{n})| + s_a + s_b + |\varrho_1(\vec{n})|$, that is, the position of the last symbol of $\varrho(\vec{n})$. Then $w[1:i] \in L_{k+3}$, while $w'[1:i] \notin L_{k+3}$. But by Lemma 4.6, we have $w[1:i] \models \phi_1 \iff w'[1:i] \models \phi_1$. This is a contradiction, so we conclude that no formula $\phi$ with depth $k$ can solve the prediction problem for $L_{k+3}$.

If $k$ is odd, the argument is the same, except with $a$ and $b$ swapped.

# D  Depth Hierarchy for $\mathsf{TL}[\overleftarrow{\#}, \overrightarrow{\#}]$

We can also obtain a strict depth hierarchy for $\mathsf{TL}[\overleftarrow{\#}, \overrightarrow{\#}]$. The key observation is that to modify Lemma 4.8 for $\mathsf{TL}[\overleftarrow{\#}, \overrightarrow{\#}]$ we use the fact that if the Parikh vector of a word is fixed we can rewrite $\overrightarrow{\#}$ in terms of $\overleftarrow{\#}$ and $\vec{n}$.

**Lemma D.1** (Cropping Lemma for $\mathsf{TL}[\overleftarrow{\#}, \overrightarrow{\#}]$, cf. Lemmas 1–2 of Behle et al., 2009). *For any formula $\phi$ of $\mathsf{TL}[\overleftarrow{\#}, \overrightarrow{\#}, \mathsf{PNP}]$ and any accommodating family of intervals $I \colon \mathbb{N}^{|\Sigma|} \to \mathcal{I}(\mathbb{N}^{|\Sigma|})$, such that $I(\vec{n}) \subseteq [\vec{0}, \vec{n}]$ and the PNPs of $\phi$ are constant on $I$, there exists an accommodating family of intervals $I' \colon \mathbb{N}^{|\Sigma|} \to \mathcal{I}(\mathbb{N}^{|\Sigma|})$ such that $I'(\vec{n}) \subseteq I(\vec{n})$ but does not stick to any side of $I(\vec{n})$ for all $\vec{n}$, and all of the minimal depth-1 subformulas (and PNPs) of $\phi$ are constant on $I'$.*

*Proof.* Let $\psi_1, \psi_2, \ldots, \psi_c$ be the minimal depth-1 subformulas of $\phi$. That is, for $\ell \in [c]$,

$$\psi_\ell = \left( \sum_{\chi \in \mathcal{L}_\ell} \lambda_\chi \overleftarrow{\#}[\chi] \right) + \left( \sum_{\chi \in \mathcal{R}_\ell} \lambda_\chi \overrightarrow{\#}[\chi] \right) \geq C_\ell.$$

For any size $\vec{s} = (s_a, s_b) \in \mathbb{N}^{|\Sigma|}$, let $\vec{s}' = (s_a + 2, s_b + 2)$. We may, since $I$ is accommodating, set $\vec{n}$ such that $I(\vec{n})$ has size at least $(2^c s'_a, 2^c s'_b)$. Then we rewrite $\psi_\ell$ as

$$\psi_\ell = \left( \sum_{\chi \in \mathcal{L}_\ell} \lambda_\chi \overleftarrow{\#}[\chi] \right) + \left( \sum_{\chi \in \mathcal{R}_\ell} \lambda_\chi (C_\chi - \overleftarrow{\mathring{\#}}[\chi]) \right) \geq C_\ell$$

$$C_\chi = \begin{cases} n_a + n_b & \text{if } \chi = \top \\ n_a & \text{if } \chi = Q_a \\ n_b & \text{if } \chi = Q_b \\ 0 & \text{if } \chi = \bot. \end{cases}$$

(Regarding the strict counting operator $\overleftarrow{\mathring{\#}}$, see Lemma A.4.) Now each $\psi_\ell$ defines a half-plane over $\overleftarrow{\#}[Q_a]$ and $\overleftarrow{\#}[Q_b]$, where $w, i \models \psi_\ell$ iff $w[1 : i]$ lands on the correct side of the half-plane. Then $\phi$ is a Boolean combination of these half-planes. Thus if $w[1 : i]$ and $w'[1 : i']$ land in the same half-planes, they will both satisfy $\phi$ or both not satisfy $\phi$.

Next, we prove that we can find an interval with size at least $\vec{s}'$ on which all the $\psi_\ell$ are constant, by induction on $c$. The base case $c = 0$ is trivial. For $c > 0$, we split the interval into four quadrants, each of size $(2^{c-1} s'_a, 2^{c-1} s'_b)$. Since a line can only intersect at most three quadrants, there is one quadrant that is completely contained in the half-plane for $\psi_c$ or completely outside. Use the inductive hypothesis on this quadrant and the remaining half-planes for $\{\psi_1, \ldots, \psi_{c-1}\}$.

Finally, shrink the interval slightly to obtain an interval $I'(\vec{n})$ of size $\vec{s}$ that does not touch any side of $I(\vec{n})$. $\qquad\square$

Then, in order to prove a depth hierarchy for $\mathsf{TL}[\overleftarrow{\#}]$, each step of reduction needs to eliminate a block on both the left and the right.

**Theorem D.2.** *Define the family of languages $D_k = L_{2k-1} = (a^+ b^+)^{k-1} a^+$. Then $D_{k+1}$ is definable in $\mathsf{TL}[\overleftarrow{\#}, \overrightarrow{\#}]_{k+1}$ but not in $\mathsf{TL}[\overleftarrow{\#}, \overrightarrow{\#}]_k$.*

*Proof.* Assume, for the sake of contradiction, that there exists some depth $k$ formula $\phi \in \mathsf{TL}[\overleftarrow{\#}, \overrightarrow{\#}]_k$ which defines $D_{k+1}$. Let $I_k(\vec{n}) = [(1, 0), \vec{n} - (1, 0)]$, $\lambda_k(\vec{n}) = \varrho_k(\vec{n}) = a$, and $\phi_k = \phi$. Note that $(\lambda_k, \varrho_k)$ is accommodating, and $\phi_k$ has no PNPs.

For $\ell = k - 1, k - 2, \ldots, 1$, we will define the following, writing $L^R$ for the *reversal* of $L$, that is, the set of reversal of strings in $L$:

1. A accommodating family of intervals $I_\ell(\vec{n}) \subseteq I_{\ell+1}(\vec{n})$.

2. An accommodating affix restriction $\lambda_\ell(\vec{n}) \in L_{k-\ell+1}$ and $\varrho_\ell(\vec{n}) \in L_{k-\ell+1}{}^R$.

3. A depth-$\ell$ formula $\phi_\ell \in \mathsf{TL}[\overleftarrow{\#}, \overrightarrow{\#}, \mathsf{PNP}]_\ell$ which defines $\lambda_\ell (D_{k+1})_{\varrho_\ell}$ and only uses PNPs which are constant over $(\lambda_\ell, \varrho_\ell)$.

We use the following iterative procedure:

1. Using Lemma D.1, find an accommodating family of intervals $I_\ell$ such that $I_\ell(\vec{n}) \subseteq I_{\ell+1}(\vec{n})$ and does not stick to any side of $I_{\ell+1}(\vec{n})$ for all $\vec{n}$, and the minimal depth-1 subformulas of $\phi_{\ell+1}$ are constant on $I$.

2. We choose an affix restriction whose middle is $I_\ell$, as follows. Let $[\vec{i}, \vec{j}] = I_{\ell+1}(\vec{n})$ and $[\vec{i'}, \vec{j'}] = I_\ell(\vec{n})$; then

$$\lambda_\ell(\vec{n}) = \begin{cases} \lambda_{\ell+1}(\vec{n}) a^{[\vec{i'}-\vec{i}]_a} b^{[\vec{i'}-\vec{i}]_b} & k - \ell \text{ odd} \\ \lambda_{\ell+1}(\vec{n}) b^{[\vec{i'}-\vec{i}]_b} a^{[\vec{i'}-\vec{i}]_a} & k - \ell \text{ even} \end{cases}$$

$$\varrho_\ell(\vec{n}) = \begin{cases} b^{[\vec{j}-\vec{j'}]_b} a^{[\vec{j}-\vec{j'}]_a} \varrho_{\ell+1}(\vec{n}) & k - \ell \text{ odd} \\ a^{[\vec{i'}-\vec{i}]_a} b^{[\vec{j}-\vec{j'}]_b} \varrho_{\ell+1}(\vec{n}) & k - \ell \text{ even.} \end{cases}$$

Because $\lambda_{\ell+1}(\vec{n}) \in L_{k-\ell-1+1}$ and $I_\ell(\vec{n})$ does not stick to any side of $I_{\ell+1}(\vec{n})$, we have that $\lambda_\ell \in L_{k-\ell+1}$. Similarly, $\varrho_\ell \in L_{k-\ell+1}^R$.

3. Using Lemma 4.9 we can find a depth-$\ell$ formula $\phi_\ell$ of $\mathsf{TL}[\overleftarrow{\#}, \overrightarrow{\#}, \mathsf{PNP}]_\ell$ that defines $\lambda_\ell(D_{k+1})_{\varrho_\ell}$ and only uses PNPs which are constant on $(\lambda_\ell, \varrho_\ell)$.

At the end of this procedure we are left with

- An accommodating affix restriction $\lambda_1(\vec{n}) \in L_k$ and $\varrho_1(\vec{n}) \in L_k^R$.

- A depth-1 formula $\phi_1 \in \mathsf{TL}[\overleftarrow{\#}, \overrightarrow{\#}, \mathsf{PNP}]_1$ which defines $\lambda_1(D_k)_{\varrho_1}$ and only uses PNPs which are constant over $(\lambda_1, \varrho_1)$.

Since $(\lambda_1, \varrho_1)$ is accommodating, choose $\vec{n}$ so that the middle has $s_a \geq 2$ occurrences of $a$ and $s_b \geq 2$ occurrences of $b$. Construct strings

$$w = \lambda_1(\vec{n}) a^{s_a} b^{s_b} \varrho_1(\vec{n})$$
$$w' = \lambda_1(\vec{n}) a^{s_a-1} b^{s_b-1} ab \varrho_1(\vec{n}).$$

The prefix $\lambda_1(\vec{n})$ and suffix $\varrho_1(\vec{n})$ both have $k$ blocks, and $\lambda_1(\vec{n})$ ends with the same letter that $\varrho_1(\vec{n})$ starts with. So $w$ has $(2k + 1)$ blocks and is therefore in $D_{k+1}$, while $w'$ has $(2k + 3)$ blocks and is therefore not in $D_{k+1}$. But by Lemma 4.6, we have $w \models \phi_1 \iff w' \models \phi_1$. This is a contradiction, so we conclude that no formula $\phi$ with depth $k$ can define $D_{k+1}$.

Finally, by Lemma 2.8, $D_{k+1} = L_{2k+1}$ is a $(2k+1)$-piecewise testable language. Thus, by Lemma 2.9, $D_{k+1}$ is definable in $\mathsf{TL}[\overleftarrow{\#}, \overrightarrow{\#}]_{k+1}$. □

# E   Equivalence of $\mathsf{TL}[\overleftarrow{\#}, \overrightarrow{\#}]$ to Other Formalisms

The logic $\mathsf{TL}[\overleftarrow{\#}, \overrightarrow{\#}]$ is equivalent to two other formalisms studied in the literature. The multiple different ways of characterizing this class of languages suggest that this is a robust class of languages.

**Definition E.1.** *The syntax of* $\widehat{\mathsf{MAJ}}_2[<]$ *is as follows:*

$$\begin{aligned} \phi ::=&\ Q_\sigma(x) \mid Q_\sigma(y) & \sigma \in \Sigma \\ &\mid x < y \mid y < x \\ &\mid \neg\phi_1 \mid \phi_1 \wedge \phi_2 \\ &\mid \widehat{\mathsf{MAJ}}_x \langle \phi_1, \dots, \phi_m \rangle \mid \widehat{\mathsf{MAJ}}_y \langle \phi_1, \dots, \phi_m \rangle & m \geq 1. \end{aligned}$$

*The semantics of formulas is defined by the relation* $w, \xi \models \phi$, *where* $\xi$ *is a partial function from variables in* $\{x, y\}$ *to truth values in* $\{0, 1\}$. *We write* $\xi[x \mapsto i]$ *for the function* $\xi'$ *such that* $\xi'(x) = i$

*and $\xi'(y) = \xi(y)$, and similarly for $\xi[y \mapsto j]$.*

$$w, \xi \models Q_\sigma(x) \quad \Longleftrightarrow \quad w_{\xi(x)} = \sigma \tag{17a}$$

$$w, \xi \models Q_\sigma(y) \quad \Longleftrightarrow \quad w_{\xi(y)} = \sigma \tag{17b}$$

$$w, \xi \models x < y \quad \Longleftrightarrow \quad \xi(x) < \xi(y) \tag{17c}$$

$$w, \xi \models y < x \quad \Longleftrightarrow \quad \xi(y) < \xi(x) \tag{17d}$$

$$w, \xi \models \neg\phi \quad \Longleftrightarrow \quad w, \xi \not\models \phi \tag{17e}$$

$$w, \xi \models \phi_1 \wedge \phi_2 \quad \Longleftrightarrow \quad w, \xi \models \phi_1 \text{ and } w, \xi \models \phi_2 \tag{17f}$$

$$w, \xi \models \widehat{\mathsf{MAJ}}_x \langle \phi_1, \ldots, \phi_m \rangle \quad \Longleftrightarrow \quad \sum_{i=1}^{|w|} \sum_{\ell=1}^{m} \mathbb{I}\big[w, \xi[x \mapsto i] \models \phi_\ell\big] > \frac{|w|m}{2} \tag{17g}$$

$$w, \xi \models \widehat{\mathsf{MAJ}}_y \langle \phi_1, \ldots, \phi_m \rangle \quad \Longleftrightarrow \quad \sum_{j=1}^{|w|} \sum_{\ell=1}^{m} \mathbb{I}\big[w, \xi[y \mapsto j] \models \phi_\ell\big] > \frac{|w|m}{2}. \tag{17h}$$

*We write $w \models \phi$ to mean $w, \emptyset \models \phi$, and we say that a closed formula $\phi$ defines the language $\mathcal{L}(\phi) = \{w \mid w \models \phi\}$.*

**Definition E.2.** *The* depth *of formulas and terms of $\widehat{\mathsf{MAJ}}_2[<]$ is defined by:*

$$\mathrm{dp}(Q_\sigma(x)) = \mathrm{dp}(Q_\sigma(y)) = 0$$
$$\mathrm{dp}(x < y) = \mathrm{dp}(y < x) = 0$$
$$\mathrm{dp}(\neg\phi) = \mathrm{dp}(\phi)$$
$$\mathrm{dp}(\phi_1 \wedge \phi_2) = \max\{\mathrm{dp}(\phi_1), \mathrm{dp}(\phi_2)\}$$
$$\mathrm{dp}(\widehat{\mathsf{MAJ}}_x \langle \phi_1, \ldots, \phi_m \rangle) = \mathrm{dp}(\widehat{\mathsf{MAJ}}_y \langle \phi_1, \ldots, \phi_m \rangle) = 1 + \max\{\mathrm{dp}(\phi_1), \ldots, \mathrm{dp}(\phi_m)\}.$$

*We write $\widehat{\mathsf{MAJ}}_2[<]_k$ for the class of all formulas $\phi$ such that $\mathrm{dp}(\phi) \le k$.*

**Lemma E.3.** *Any formula $\phi$ of $\widehat{\mathsf{MAJ}}_2[<]$ that uses $\forall$ or $\exists$ can be converted into a formula that does not use $\forall$ or $\exists$, defines the same language as $\phi$, and has the same depth as $\phi$.*

*Proof.* These quantifiers can be rewritten equivalently using $\widehat{\mathsf{MAJ}}$:

$$\exists x[\phi] \equiv \widehat{\mathsf{MAJ}}_x \langle \phi, \top \rangle$$
$$\forall x[\phi] \equiv \neg\widehat{\mathsf{MAJ}}_x \langle \neg\phi, \top \rangle. \qquad \square$$

Now, we show the equivalence of $\mathsf{TL}[\overleftarrow{\#}, \overrightarrow{\#}]$ with $\widehat{\mathsf{MAJ}}_2[<]$. First, we show how to translate $\mathsf{TL}[\overleftarrow{\#}, \overrightarrow{\#}]$ to $\widehat{\mathsf{MAJ}}_2[<]$ (Theorem E.4) and then how to translate $\widehat{\mathsf{MAJ}}_2[<]$ to $\mathsf{TL}[\overleftarrow{\#}, \overrightarrow{\#}]$ (Theorem E.6).

**Theorem E.4.** *Let $\phi$ be a formula of $\mathsf{TL}[\overleftarrow{\#}, \overrightarrow{\#}]_k$. Then there exists a $\widehat{\mathsf{MAJ}}_2[<]_k$ formula $\phi'(x)$ with one free variable such that $w, i \models \phi \iff w, x = i \models \phi'(x)$ for all $w$ and $1 \le i \le |w|$.*

*Proof.* We define a transformation $\mathcal{M}_x \llbracket \cdot \rrbracket$ from formulas of $\mathsf{TL}[\overleftarrow{\#}, \overrightarrow{\#}]$ to formulas of $\widehat{\mathsf{MAJ}}_2[<]$ with one free variable $x$:

$$\mathcal{M}_x \llbracket Q_\sigma \rrbracket = Q_\sigma(x)$$
$$\mathcal{M}_x \llbracket \neg\phi \rrbracket = \neg\mathcal{M}_x \llbracket \phi \rrbracket$$
$$\mathcal{M}_x \llbracket \phi_1 \wedge \phi_2 \rrbracket = \mathcal{M}_x \llbracket \phi_1 \rrbracket \wedge \mathcal{M}_x \llbracket \phi_2 \rrbracket.$$

Any comparison formula can be written in the form

$$\sum_{\ell=1}^{m} t_\ell - \sum_{\ell=1}^{m'} t'_\ell > 0$$

where $t_\ell$ and $t'_\ell$ are terms. Since this tests whether the sum is greater than 0, whereas the $\widehat{\mathsf{MAJ}}$ quantifier tests whether the sum is greater than half of its maximum possible value, we need to pad

the positive terms ($t_\ell$) with an equal number of trivially true formulas. Similarly, we need to pad the negative terms ($t'_\ell$) with an equal number of trivially false formulas.

$$\mathcal{M}_x \left[\!\!\left[ \sum_{\ell=1}^{m} t_\ell - \sum_{\ell=1}^{m'} t'_\ell > 0 \right]\!\!\right] = \widehat{\mathsf{MAJ}}_y \langle \mathcal{F}_x [\![ t_1 ]\!] , \top , \ldots , \mathcal{F}_x [\![ t_m ]\!] , \top , \neg \mathcal{F}_x [\![ t'_1 ]\!] , \bot , \ldots , \neg \mathcal{F}_x [\![ t'_{m'} ]\!] , \bot \rangle$$

(18a)

$$\mathcal{F}_x \left[\!\!\left[ \overleftarrow{\#} [\phi] \right]\!\!\right] = (y \leq x \wedge \mathcal{M}_y [\![ \phi ]\!]) \tag{18b}$$

$$\mathcal{F}_x \left[\!\!\left[ \overrightarrow{\#} [\phi] \right]\!\!\right] = (y \geq x \wedge \mathcal{M}_y [\![ \phi ]\!]) \tag{18c}$$

$$\mathcal{F}_x [\![ \#[\phi] ]\!] = \mathcal{M}_y [\![ \phi ]\!] \tag{18d}$$

$$\mathcal{F}_x [\![ 1 ]\!] = (y = x). \tag{18e}$$

To see why this works, we can show by induction that for all strings $w$, assignments $\xi$, formulas $\phi$, and terms $t$, both of the following hold:

$$w, \xi \models \mathcal{M}_x [\![ \phi ]\!] \iff w, \xi(x) \models \phi \tag{19a}$$

$$\sum_{j=1}^{|w|} \mathbb{I}\big[ w, \xi[y \mapsto j] \models \mathcal{F}_x [\![ t ]\!] \big] = t^{w, \xi(x)}. \tag{19b}$$

The interesting case is

$$w, \xi \models \mathcal{M}_x \left[\!\!\left[ \sum_{\ell=1}^{m} t_\ell - \sum_{\ell=1}^{m'} t'_\ell > 0 \right]\!\!\right]$$

$$\overset{(18a)}{\iff} w, \xi \models \widehat{\mathsf{MAJ}}_y \langle \mathcal{F}_x [\![ t_1 ]\!] , \top , \ldots , \mathcal{F}_x [\![ t_m ]\!] , \top , \neg \mathcal{F}_x [\![ t'_1 ]\!] , \bot , \ldots , \neg \mathcal{F}_x [\![ t'_{m'} ]\!] , \bot \rangle$$

$$\overset{(17h)}{\iff} \sum_{j=1}^{|w|} \left( \sum_{\ell=1}^{m} \Big( \mathbb{I}[w, \xi[y \mapsto j] \models \mathcal{F}_x [\![ t_\ell ]\!]] + \mathbb{I}[\top] \Big) \right.$$

$$\left. + \sum_{\ell=1}^{m'} \Big( 1 - \mathbb{I}[w, \xi[y \mapsto j] \models \mathcal{F}_x [\![ t'_\ell ]\!]] + \mathbb{I}[\bot] \Big) \right) > |w|(m + m')$$

$$\iff \sum_{j=1}^{|w|} \left( \sum_{\ell=1}^{m} \mathbb{I}[w, \xi[y \mapsto j] \models \mathcal{F}_x [\![ t_\ell ]\!]] - \sum_{\ell=1}^{m'} \mathbb{I}[w, \xi[y \mapsto j] \models \mathcal{F}_x [\![ t'_\ell ]\!]] \right) > 0$$

$$\overset{(19b)}{\iff} \sum_{\ell=1}^{m} (t_\ell)^{w, \xi(x)} - \sum_{\ell=1}^{m'} (t_\ell)^{w, \xi(x)} > 0$$

$$\overset{(4c)}{\iff} w, \xi(x) \models \sum_{\ell=1}^{m} t_\ell - \sum_{\ell=1}^{m'} t_\ell > 0.$$

Observe that a formula of the form $\mathcal{M}_x [\![ \psi ]\!]$ may only have free variable $x$, because in Eq. (18a), $\widehat{\mathsf{MAJ}}_y$ binds $y$. □

In the special case of a comparison formula of # terms, the resulting $\widehat{\mathsf{MAJ}}_2[<]$ formula will be closed.

**Proposition E.5.** *Let* $\mathcal{M}_x [\![ \cdot ]\!]$ *be as in Theorem E.4. If* $\phi$ *is of the form* $\#[\psi] > 0$, *then* $\mathcal{M}_x [\![ \phi ]\!]$ *is closed.*

*Proof.* Recall that a formula of the form $\mathcal{M}_x [\![ \psi ]\!]$ may only have free variable $x$, because in Eq. (18a), $\widehat{\mathsf{MAJ}}_y$ binds $y$. But the special case of $\mathcal{M}_x [\![ \#[\psi] > 0 ]\!]$ is closed, because $\mathcal{F}_x [\![ \#[\psi] ]\!] = \mathcal{M}_y [\![ \psi ]\!]$ (Eq. (18d)) only has free variable $y$, and Eq. (18a), $\widehat{\mathsf{MAJ}}_y$ binds $y$. □

**Theorem E.6.** *Let* $\phi(x)$ *be a formula of* $\widehat{\mathsf{MAJ}}_2[<]_k$ *with one free variable* $x$. *Then there exists a* $\mathsf{TL}[\overleftarrow{\#}, \overrightarrow{\#}]_k$ *formula* $\phi'(x)$ *such that for all* $w$ *and all* $i \in [|w|]$, *we have* $w, x = i \models \phi(x) \iff w, i \models \phi'$.

*Proof.* We define a transformation $\mathcal{T}_x$ that transforms a formula of $\widehat{\mathsf{MAJ}}_2[<]$ with free variable $x$ into a formula of $\mathsf{TL}[\overleftarrow{\#}, \overrightarrow{\#}]$.

$$\mathcal{T}_x \llbracket Q_\sigma(x) \rrbracket = Q_\sigma$$
$$\mathcal{T}_x \llbracket \neg\phi(x) \rrbracket = \neg\mathcal{T}_x \llbracket \phi(x) \rrbracket$$
$$\mathcal{T}_x \llbracket \phi_1(x) \wedge \phi_2(x) \rrbracket = \mathcal{T}_x \llbracket \phi_1(x) \rrbracket \wedge \mathcal{T}_x \llbracket \phi_2(x) \rrbracket$$
$$\mathcal{T}_x \llbracket \widehat{\mathsf{MAJ}}_y \langle \phi_1(x,y), \ldots, \phi_m(x,y) \rangle \rrbracket = \sum_{\ell \in [m]} C_y \llbracket \phi_\ell(x,y) \rrbracket > \sum_{\ell \in [m]} C_y \llbracket \neg\phi_\ell(x,y) \rrbracket .$$

The transformation $\mathcal{T}_y$ is defined similarly.

The transformation $C_y \llbracket \psi(x,y) \rrbracket$, in turn, can be read as "count the number of positions $y$ that make $\psi(x,y)$ true." Without loss of generality, assume that $\psi$ is in full disjunctive normal form, that is, $\psi = \bigvee_{\ell \in [m']} \psi_\ell$ and at most one of the $\psi_\ell$ can be true at the same time. Then we define

$$C_y \left\llbracket \bigvee_{\ell \in [m']} \psi_\ell(x) \right\rrbracket = \sum_{\ell \in [m']} C_y \llbracket \psi_\ell(x) \rrbracket .$$

Each of the $\psi_\ell$ can be written as a conjunction of literals with free variable $x$, literals with free variable $y$, and possibly a comparison $x < y$, $x \le y$, $y \le x$, or $y < x$. Then we define

$$C_y \llbracket \psi_{\ell 1}(x) \wedge \psi_{\ell 2}(y) \wedge x < y \rrbracket = \psi_{\ell 1} \; ? \; \overset{\circ}{\overrightarrow{\#}} \left[ \mathcal{T}_y \llbracket \psi_{\ell 2}(y) \rrbracket \right] \; : 0$$
$$C_y \llbracket \psi_{\ell 1}(x) \wedge \psi_{\ell 2}(y) \wedge x \le y \rrbracket = \psi_{\ell 1} \; ? \; \overrightarrow{\#} \left[ \mathcal{T}_y \llbracket \psi_{\ell 2}(y) \rrbracket \right] \; : 0$$
$$C_y \llbracket \psi_{\ell 1}(x) \wedge \psi_{\ell 2}(y) \wedge y < x \rrbracket = \psi_{\ell 1} \; ? \; \overset{\circ}{\overleftarrow{\#}} \left[ \mathcal{T}_y \llbracket \psi_{\ell 2}(y) \rrbracket \right] \; : 0$$
$$C_y \llbracket \psi_{\ell 1}(x) \wedge \psi_{\ell 2}(y) \wedge y \le x \rrbracket = \psi_{\ell 1} \; ? \; \overleftarrow{\#} \left[ \mathcal{T}_y \llbracket \psi_{\ell 2}(y) \rrbracket \right] \; : 0$$
$$C_y \llbracket \psi_{\ell 1}(x) \wedge \psi_{\ell 2}(y) \rrbracket = \psi_{\ell 1} \; ? \; \# \left[ \mathcal{T}_y \llbracket \psi_{\ell 2}(y) \rrbracket \right] \; : 0.$$

The transformation $C_x$ is defined similarly. (Regarding the strict counting operators $\overset{\circ}{\overleftarrow{\#}}$ and $\overset{\circ}{\overrightarrow{\#}}$, see Lemma A.4.) $\qquad\square$

**Theorem E.7.** $\mathcal{L}(\mathsf{TL}[\overleftarrow{\#}, \overrightarrow{\#}]) = \mathcal{L}(\widehat{\mathsf{MAJ}}_2[<]) = \mathcal{L}(\mathsf{FO}[<]\text{-uniform } \mathsf{LTC}^0)$. *Furthermore, for all* $k \ge 0$ *we have* $\mathsf{TL}[\overleftarrow{\#}, \overrightarrow{\#}]_k \subseteq \widehat{\mathsf{MAJ}}_2[<]_{k+1}$ *and* $\widehat{\mathsf{MAJ}}_2[<]_k \subseteq \mathsf{TL}[\overleftarrow{\#}, \overrightarrow{\#}]_k$.

*Proof.* The equivalence $\mathcal{L}(\widehat{\mathsf{MAJ}}_2[<]) = \mathcal{L}(\mathsf{FO}[<]\text{-uniform } \mathsf{LTC}^0)$ was shown by Krebs (2008, Theorem 4.33).

We will show the following:

- $\mathcal{L}(\mathsf{TL}[\overleftarrow{\#}, \overrightarrow{\#}]_k) \subseteq \mathcal{L}(\widehat{\mathsf{MAJ}}_2[<]_{k+1})$ using Theorem E.4.

- $\mathcal{L}(\widehat{\mathsf{MAJ}}_2[<]_k) \subseteq \mathcal{L}(\mathsf{TL}[\overleftarrow{\#}, \overrightarrow{\#}]_k)$ using Theorem E.6.

If $\phi$ is a formula of $\mathsf{TL}[\overleftarrow{\#}, \overrightarrow{\#}]_k$, then by Theorem E.4, there is an equivalent $\widehat{\mathsf{MAJ}}_2[<]_k$ formula $\phi'(x)$. This, in turn, is equivalent to the following closed formula:

$$\phi'' = \exists x.(\neg\exists y.y > x) \wedge \phi'(x).$$

This accounts for the end-satisfaction of $\mathsf{TL}[\overleftarrow{\#}, \overrightarrow{\#}]$ formulas, but adds a level of depth to the $\widehat{\mathsf{MAJ}}_2[<]$ formula.

Conversely, if $\phi$ is a closed formula of $\widehat{\mathsf{MAJ}}_2[<]_k$, we may think of it as having one free variable $x$, so by Theorem E.6 below, there is a $\mathsf{TL}[\overleftarrow{\#}, \overrightarrow{\#}]_k$ formula equivalent to $\phi$. $\qquad\square$

This theorem combined with our Theorem D.2 implies the following answer to an open question.

**Corollary E.8.** *The circuit depth hierarchy for* $\mathsf{FO}[<]\text{-uniform } \mathsf{LTC}^0$ *circuits is strict.*

*Proof.* By Theorem D.2 and Theorem E.7 we know that $D_{k+1} \notin \mathcal{L}(\widehat{\mathsf{MAJ}}_2[<]_k)$. On the other hand, let $\phi_k$ be the $\mathsf{TL}[\overleftarrow{\#}, \overrightarrow{\#}]_k$ formula given by Lemma 2.9 for $D_k$. Then apply Theorem E.4 to get a formula $\phi'_k$ of $\widehat{\mathsf{MAJ}}_2[<]_k$ that defines $D_k$. Since $\phi_k$ is of the form $\#[\psi] > 0$, by Proposition E.5, $\phi'_k$ is closed. Thus, the depth hierarchy for $\widehat{\mathsf{MAJ}}_2[<]$ is strict.

By Theorem 3 of Behle et al. (2013), FO[<]-uniform LTC$^0$ circuits form a hierarchy in the circuit depth iff $\widehat{\mathsf{MAJ}}_2[<]$ formulas form a hierarchy in the quantifier depth. Their theorem states that there exists a constant $c$ such that a circuit of depth $k$ can be expressed as a formula of depth $k + c$, and a formula of depth $k$ can be expressed as a circuit of depth $ck$. □

# F  Position Encodings

In this section, we extend the results of Sections 3 and 4 to handle position encodings. On the logic side, we extend $\mathsf{TL}[\overleftarrow{\#}]$ with new predicates $\mathsf{MOD}^r_m$, which test whether the position is congruent to $r$ modulo $m$, and a new operator $\mathbf{Y}\phi$, which tests whether $\phi$ is true at the previous position. On the transformer side, we consider the original sinusoidal position encodings, as well as RoPE (Su et al., 2024) and ALiBi (Press et al., 2022).

## F.1  $\mathsf{TL}[\overleftarrow{\#}, \mathbf{Y}, \mathsf{MOD}]$

We extend $\mathsf{TL}[\overleftarrow{\#}]$ to a more expressive logic, $\mathsf{TL}[\overleftarrow{\#}, \mathbf{Y}, \mathsf{MOD}]$ (which is equivalent to $\mathsf{C\text{-}RASP}[\mathsf{local}, \mathsf{periodic}]$ of Huang et al. (2025)). The new syntax rules are:

$$\phi ::= \mathbf{Y}\phi_1 \mid \mathsf{MOD}^r_m$$

The semantics of the extensions are defined as follows:

$$w, i \models \mathbf{Y}\phi \quad \Longleftrightarrow \quad w, (i - 1) \models \phi \text{ and } i > 1 \tag{20a}$$
$$w, i \models \mathsf{MOD}^r_m \quad \Longleftrightarrow \quad i \equiv r \pmod{m}. \tag{20b}$$

## F.2  Depth Hierarchy

To prove a strict depth hierarchy for $\mathsf{TL}[\overleftarrow{\#}, \mathbf{Y}, \mathsf{MOD}]$, we first we define an intermediate step to simplify the construction. A formula is in $\mathbf{Y}$-*normal form* if $\mathbf{Y}$ only appears around atomic formulas. That is, the set of formulas in $\mathbf{Y}$-normal form is defined by the following grammar.

$$\phi ::= t_1 < t_2 \mid \neg\phi_1 \mid \phi_1 \wedge \phi_2 \mid \psi$$
$$\psi ::= Q_\sigma \mid \mathsf{MOD}^r_m \mid \mathbf{Y}\psi_1$$
$$t ::= \overleftarrow{\#}[\phi] \mid t_1 + t_2 \mid 1$$

Below, we will use the shorthand, for any $c \geq 0$,

$$\mathbf{Y}^c \phi = \underbrace{\mathbf{Y} \cdots \mathbf{Y}}_{c \text{ times}} \phi.$$

**Lemma F.1.** *For every formula $\phi$ of $\mathsf{TL}[\overleftarrow{\#}, \mathbf{Y}, \mathsf{MOD}]_k$ there exists a formula $\phi'$ of $\mathsf{TL}[\overleftarrow{\#}, \mathbf{Y}, \mathsf{MOD}]_k$ such that $w, i \models \phi \iff w, i \models \phi'$ for all $w \in \Sigma^*$, and $\phi'$ is in $\mathbf{Y}$-normal form.*

*Proof.* Define a transformation $\mathcal{N}^c[\![\cdot]\!]$, where $c \geq 0$, applying to both formulas and terms, that pushes $\mathbf{Y}$'s inwards. The superscript $c$ keeps track of how many $\mathbf{Y}$'s are being pushed. For any formula $\phi$, we have $w, i \models \phi \iff w, i - c \models \mathcal{N}^c[\![\phi]\!]$, and for any term $t$, we have $t^{w,i} = \mathcal{N}^c[\![t]\!]^{w,i-c}$.

$$\mathcal{N}^c[\![Q_\sigma]\!] = \mathbf{Y}^c Q_\sigma$$
$$\mathcal{N}^c[\![\mathsf{MOD}^r_m]\!] = \mathbf{Y}^c \mathsf{MOD}^r_m$$
$$\mathcal{N}^c[\![\neg\phi]\!] = \neg\mathcal{N}^c[\![\phi]\!]$$
$$\mathcal{N}^c[\![\phi_1 \wedge \phi_2]\!] = \mathcal{N}^c[\![\phi_1]\!] \wedge \mathcal{N}^c[\![\phi_2]\!]$$
$$\mathcal{N}^c[\![t_1 < t_2]\!] = \mathcal{N}^c[\![t_1]\!] < \mathcal{N}^c[\![t_2]\!]$$
$$\mathcal{N}^c[\![\mathbf{Y}\phi]\!] = \mathcal{N}^{c+1}[\![\phi]\!]$$
$$\mathcal{N}^c[\![\overleftarrow{\#}[\phi]]\!] = \overleftarrow{\#}[\mathcal{N}^c[\![\phi]\!]]$$
$$\mathcal{N}^c[\![t_1 + t_2]\!] = \mathcal{N}^c[\![t_1]\!] + \mathcal{N}^c[\![t_2]\!]$$
$$\mathcal{N}^c[\![1]\!] = 1.$$

Now for any formula $\phi$ of $\mathsf{TL}[\overleftarrow{\#}, \mathbf{Y}, \mathsf{MOD}]_k$, it is easily verified that $\phi' = \mathcal{N}^0[\![\phi]\!]$ is of the same depth and in the desired normal form. □

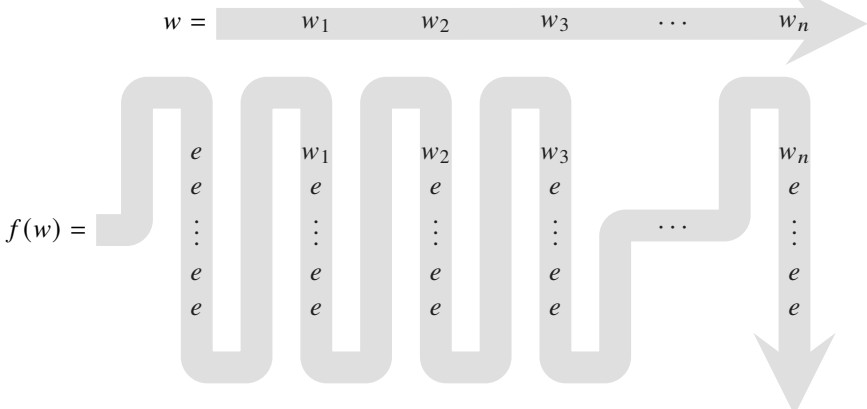

Figure 5: A set of $x$ many formulas on a string $w$ of length $n$ can simulate a formula $\phi$ on a string $f(w)$ of length $x(|w| + 1)$.

**Lemma F.2.** *Let $\phi$ be a formula of $\mathsf{TL}[\overleftarrow{\#}, \mathbf{Y}, \mathsf{MOD}]_k$ over alphabet $\Sigma \cup \{e\}$ for $e \notin \Sigma$. There exists a formula $\phi'$ of $\mathsf{TL}[\overleftarrow{\#}]_k$ and a mapping (for some $x \geq 1$)*

$$f \colon \Sigma^* \to (\Sigma \cup \{e\})^*$$
$$w_1 w_2 \cdots w_n \mapsto e^x w_1 e^{x-1} w_2 e^{x-1} \cdots w_n e^{x-1}$$

*such that for all $w \in \Sigma^*$, $f(w) \models \phi \iff w \models \phi'$.*

*Proof.* First, let $M$ be the least common multiple of all moduli used in $\phi$ (or $M = 1$ if there are none), and let $Y$ be the $\mathbf{Y}$-depth of $\phi$. Set $x = M(Y + 1)$. This ensures that $x \equiv 0 \bmod m$ for any modulus $m$, and $x > Y$, which are important conditions for the following proof.

Intuitively, the technical challenge here is that in $w$ we can index only $|w|$ many positions, but to simulate a formula over $f(w)$, we need to simulate a procedure which can index $x(|w| + 1)$ many positions. We address this by defining a transformation $\mathcal{T}_y \llbracket \cdot \rrbracket$, for $y \in [x]$, from formulas of $\mathsf{TL}[\overleftarrow{\#}, \mathbf{Y}, \mathsf{MOD}]$ to $\mathsf{TL}[\overleftarrow{\#}]$. At position $i$, $\mathcal{T}_y \llbracket \phi \rrbracket$ simulates $\phi$ at position $(xi + y)$ (and similarly for terms $t$). That is, for all $\phi$ and $t$, and for all strings $w$ and $i \in [|w|]$,

$$f(w), xi + y \models \phi \iff w, i \models \mathcal{T}_y \llbracket \phi \rrbracket \qquad\qquad t^{f(w), xi+y} = \mathcal{T}_y \llbracket t \rrbracket^{w,i} . \qquad (21)$$

The first $x$ positions in $f(w)$ are not simulated; they are dealt with specially below. In the end, Eq. (21) will ensure that $f(w) \models \phi \iff w \models \phi_x$. Intuitively, this construction "stores $f(w_i)$ vertically" at each symbol $w_i$ in $w$, and thus we can simulate, in place, the value of $\phi$ at each position of $f(w_i)$. We can visualize $w = w_1 w_2 w_3 \cdots w_n$ and $f(w)$ as in Fig. 5, with each $f(w_i)$ viewed as a vertical column of symbols in an array. We can read $w$ left-to-right, and read $f(w)$ up-to-down within each column, and left-to-right across all columns.

Without loss of generality, we assume by Lemma F.1 that $\phi$ is in **Y**-normal form. Then we define the following transformation:

$$\mathcal{T}_y \left[\!\left[ \mathbf{Y}^c Q_\sigma \right]\!\right] = \begin{cases} Q_\sigma & y - c = 1 \\ \bot & \text{otherwise} \end{cases} \qquad \sigma \in \Sigma$$

$$\mathcal{T}_y \left[\!\left[ \mathbf{Y}^c Q_e \right]\!\right] = \begin{cases} \bot & y - c = 1 \\ \top & \text{otherwise} \end{cases}$$

$$\mathcal{T}_y \left[\!\left[ \mathbf{Y}^c \mathsf{MOD}_m^r \right]\!\right] = \begin{cases} \top & y - c \equiv r \bmod m \\ \bot & \text{otherwise} \end{cases}$$

$$\mathcal{T}_y \left[\!\left[ \neg \phi \right]\!\right] = \neg \mathcal{T}_y \left[\!\left[ \phi \right]\!\right]$$

$$\mathcal{T}_y \left[\!\left[ \phi_1 \wedge \phi_2 \right]\!\right] = \mathcal{T}_y \left[\!\left[ \phi_1 \right]\!\right] \wedge \mathcal{T}_y \left[\!\left[ \phi_2 \right]\!\right]$$

$$\mathcal{T}_y \left[\!\left[ t_1 < t_2 \right]\!\right] = \mathcal{T}_y \left[\!\left[ t_1 \right]\!\right] < \mathcal{T}_y \left[\!\left[ t_2 \right]\!\right]$$

$$\mathcal{T}_y \left[\!\left[ \overleftarrow{\#}[\phi] \right]\!\right] = \left( \sum_{y' \in [x]} e^x, y' \models \phi \,?\, 1 : 0 \right) + \left( \sum_{y' \in [x]} \overset{\circ}{\overleftarrow{\#}} [\mathcal{T}_{y'} \left[\!\left[ \phi \right]\!\right]] \right) + \left( \sum_{y' \in [y]} \mathcal{T}_{y'} \left[\!\left[ \phi \right]\!\right] \,?\, 1 : 0 \right)$$

$$\mathcal{T}_y \left[\!\left[ t_1 + t_2 \right]\!\right] = \mathcal{T}_y \left[\!\left[ t_1 \right]\!\right] + \mathcal{T}_y \left[\!\left[ t_2 \right]\!\right]$$

$$\mathcal{T}_y \left[\!\left[ 1 \right]\!\right] = 1.$$

We prove Eq. (21) by induction on the structure of $\phi$.

- If $\phi = \mathbf{Y}^c Q_\sigma$ for $\sigma \in \Sigma \cup \{e\}$: If $y - c = 1$, then

$$f(w), xi + y \models \mathbf{Y}^c Q_\sigma \iff f(w)_{xi} = \sigma \iff w_i = \sigma.$$

But if $y - c \neq 1$, then

$$f(w), xi + y \models \mathbf{Y}^c Q_\sigma \iff f(w)_{xi+y-c} = \sigma \iff e = \sigma.$$

In either case, and whether $\sigma \in \Sigma$ or $\sigma = e$, Eq. (21) holds.

- If $\phi = \mathbf{Y}^c \mathsf{MOD}_m^r$: Because $x$ is a multiple of every $m$, we have

$$\begin{aligned} f(w), xi + y \models \mathbf{Y}^c \mathsf{MOD}_m^r &\iff f(w), xi + y - c \models \mathsf{MOD}_m^r \\ &\iff y - c \equiv r \bmod m \\ &\iff w, i \models \mathcal{T}_y \left[\!\left[ \mathbf{Y}^c \mathsf{MOD}_m^r \right]\!\right]. \end{aligned}$$

- If $t = \overleftarrow{\#}[\phi]$: We split the count into three parts,

$$\begin{aligned} (\overleftarrow{\#}[\phi])^{f(w), xi+y} &= |\{ j \in [xi + y] \mid f(w), j \models \phi \}| \\ &= |\{ j \in [1, x] \mid f(w), j \models \phi \}| \\ &\quad + |\{ j \in [x + 1, xi] \mid f(w), j \models \phi \}| \\ &\quad + |\{ j \in [xi + 1, xi + y] \mid f(w), j \models \phi \}| \end{aligned}$$

In relation to Fig. 5, the first term sums the first column, the second term sums the columns corresponding to $w_1 \cdots w_{n-1}$, and the third term sums the last column (corresponding to

$w_n$). Taking these three terms one at a time:

$$|\{j \in [1,x] \mid f(w), j \models \phi\}| = |\{j \in [1,x] \mid e^x, j \models \phi\}|$$

$$= \left( \sum_{y' \in [x]} e^x, y' \models \phi \; ? \; 1 : 0 \right)^{w,i}$$

$$|\{j \in [x+1, xi] \mid f(w), j \models \phi\}| = \sum_{y' \in [x]} |\{i' \in [1, i-1] \mid f(w), xi' + y' \models \phi\}|$$

$$\stackrel{\text{ind. hyp.}}{=} \sum_{y' \in [x]} \left| \{i' \in [1, i-1] \mid w, i' \models \mathcal{T}_{y'} \llbracket \phi \rrbracket \} \right|$$

$$= \left( \sum_{y' \in [x]} \overleftarrow{\#} [\mathcal{T}_{y'} \llbracket \phi \rrbracket] \right)^{w,i}$$

$$|\{j \in [xi+1, xi+y] \mid f(w), j \models \phi\}| = |\{y' \in [1, y] \mid f(w), xi + y' \models \phi\}|$$

$$\stackrel{\text{ind. hyp.}}{=} |\{y' \in [1, y] \mid w, i \models \mathcal{T}_{y'} \llbracket \phi \rrbracket \}|$$

$$= \left( \sum_{y' \in [y]} \mathcal{T}_{y'} \llbracket \phi \rrbracket \; ? \; 1 : 0 \right).$$

- The remaining cases are straightforward.

By construction, $\mathcal{T}_x \llbracket \phi \rrbracket$ has the same depth as $\phi$. Setting $\phi' = \mathcal{T}_x \llbracket \phi \rrbracket$ completes the proof. $\qquad\square$

Finally, we show the depth separation for $\mathsf{TL}[\overleftarrow{\#}, \mathbf{Y}, \mathsf{MOD}]$. Let $E_k$ be the language formed by allowing unlimited insertions of $e$ anywhere into strings of $L_k$. In other words, $E_k = \mathrm{del}^{-1}(L_k)$, where $\mathrm{del} \colon (\Sigma \cup \{e\})^* \to \Sigma^*$ is the string homomorphism given by $\mathrm{del}(\sigma) = \sigma$ for $\sigma \in \Sigma$ and $\mathrm{del}(e) = \epsilon$.

**Theorem F.3.** *Let $k > 0$. The language $E_{k+1}$ is definable in $\mathsf{TL}[\overleftarrow{\#}, \mathbf{Y}, \mathsf{MOD}]_{k+1}$ but not in $\mathsf{TL}[\overleftarrow{\#}, \mathbf{Y}, \mathsf{MOD}]_k$.*

*Proof.* Suppose $\phi \in \mathsf{TL}[\overleftarrow{\#}, \mathbf{Y}, \mathsf{MOD}]_{k+1}$ defines $E_{k+1}$. By Lemma F.2, there is a string mapping $f$ and a formula $\phi' \in \mathsf{TL}[\overleftarrow{\#}]_k$ such that for all $w \in \Sigma^*$, $f(w) \models \phi \iff w \models \phi'$. However, this implies that $w \models \phi' \iff w \in L_{k+1}$, which contradicts Theorem 4.10. $\qquad\square$

## F.3 Sinusoidal position encoding

The original definition of transformers (Vaswani et al., 2017) used sinusoidal position encoding, which redefines Eq. (5) as follows. Assume $d$ is even. Define the rotation matrix

$$R(\vec{\theta}) = \mathrm{round} \begin{bmatrix} \cos\theta_1 & -\sin\theta_1 & 0 & 0 & \cdots & 0 & 0 \\ \sin\theta_1 & \cos\theta_1 & 0 & 0 & \cdots & 0 & 0 \\ 0 & 0 & \cos\theta_2 & -\sin\theta_2 & \cdots & 0 & 0 \\ 0 & 0 & \sin\theta_2 & \cos\theta_2 & \cdots & 0 & 0 \\ \vdots & \vdots & \vdots & \vdots & \ddots & \vdots & \vdots \\ 0 & 0 & 0 & 0 & \cdots & \cos\theta_{d/2} & -\sin\theta_{d/2} \\ 0 & 0 & 0 & 0 & \cdots & \sin\theta_{d/2} & \cos\theta_{d/2} \end{bmatrix} \tag{22}$$

where for $c \in [d/2]$, $\theta_c = 1000^{-2(c-1)/d}$. Then

$$\mathbf{h}_i^{(0)}(w) = E(w_i) + R(\vec{\theta})^{i-1} \begin{bmatrix} \sin 0 \\ \cos 0 \\ \vdots \\ \sin 0 \\ \cos 0 \end{bmatrix}. \tag{23}$$

For the rest of this section, let us assume that in a sinusoidal position encoding, all the angles are $\vec{\theta}$ are rational (that is, rational multiples of $\pi$), so that the encodings are periodic. Then we can extend Theorem 3.1 to transformers with sinusoidal positional encoding and TL[$\overleftarrow{\#}$, MOD], using exactly the same technique as Yang et al. (2024, Cor. 8).

**Theorem F.4.** *A language $L$ is defined by a formula of* TL[$\overleftarrow{\#}$, MOD] *of depth $k \geq 1$ if and only if* <BOS> $\cdot$ *$L$ is recognized by a depth-$k$ fixed-precision transformer with sinusoidal positional encoding.*

**Theorem F.5.** *A depth-$(k+1)$ fixed-precision transformer with sinusoidal positional encoding can recognize $E_{k+1}$, but no depth-$k$ fixed-precision transformer with sinusoidal positional encoding can.*

*Proof.* Firstly, $E_{k+1}$ is definable by a fixed-precision transformer of depth $k+1$ even without sinusoidal positional encodings. Secondly, by Theorem F.4, every language definable by a fixed-precision transformer with RoPE is definable in TL[$\overleftarrow{\#}$, MOD]$_k$, but by Theorem F.3, $E_{k+1}$ is not definable in TL[$\overleftarrow{\#}$, MOD]$_k$. □

### F.4 RoPE

Rotary Positional Embedding or RoPE (Su et al., 2024) is currently the *de facto* standard method for incorporating positional information in transformers (e.g., Mesnard et al., 2024). It modifies Eq. (9) as follows:

$$\mathbf{s}_{ij}^{(\ell)}(w) = R(\vec{\theta})^i \mathbf{q}_i^{(\ell)}(w) \cdot R(\vec{\theta})^j \mathbf{k}_j^{(\ell)}(w) \tag{24}$$

where $R$ is as in Eq. (22).

Again, let us assume that the angles in $\vec{\theta}$ are rational, so that the transformation $R(\vec{\theta})^i$ is periodic in $i$. This ultimately allows simulation using MOD.

**Proposition F.6.** *Let $T$ be a depth-$k$ fixed-precision transformer with RoPEs. There exists a depth $k$-formula of* TL[$\overleftarrow{\#}$, MOD] *that simulates $T$.*

*Proof sketch.* This is a straightforward adaptation of Proposition B.6. The rotation matrices $R(\vec{\theta})^i$ and $R(\vec{\theta})^j$ can be computed in fixed-precision using Lemma B.5 and MOD, as in the proof of Yang et al. (2024, Cor. 8). The attention scores $\mathbf{s}_{ij} = R(\vec{\theta})^i \mathbf{q}_i \cdot R(\vec{\theta})^j \mathbf{k}_j$ can be computed using Lemma B.5 and the trick of enumerating all possible queries, as in the proof of Proposition B.6. □

**Theorem F.7.** *A depth-$(k+1)$ fixed-precision transformer with RoPE can recognize $E_{k+1}$, but no depth-$k$ fixed-precision transformer with RoPE can.*

*Proof.* Firstly, $E_{k+1}$ is definable by a fixed-precision transformer of depth $k+1$ even without RoPE. Secondly, by Proposition F.6, every language definable by a fixed-precision transformer with RoPE is definable in TL[$\overleftarrow{\#}$, MOD]$_k$, but by Theorem F.3, $E_{k+1}$ is not definable in TL[$\overleftarrow{\#}$, MOD]$_k$. □

### F.5 ALiBi

ALiBi stands for Attention with Linear Bias, introduced by Press et al. (2022) as a method for improving length generalization in transformers. It decreases the attention scores ($\mathbf{s}_{ij}$) by an amount that scales linearly with the distance between the key and query positions $i$ and $j$. That is, it modifies Eq. (9) to:

$$\mathbf{s}_{ij}^{(\ell)}(w) = \mathbf{q}_i^{(\ell)}(w) \cdot \mathbf{k}_j^{(\ell)}(w) - a \cdot (i - j). \tag{25}$$

First, we note that there is some distance beyond which ALiBi rounds the attention score to 0. This ultimately allows simulation using $\mathbf{Y}$, as the following shows.

**Lemma F.8.** *Let $\mathbb{F}$ be a fixed-precision representation and let $a > 0$. There exists $\Delta_a$ such that for all $j \leq i - \Delta_a$ and $x \in \mathbb{F}$ we have that $\mathrm{round}(\exp(x - a(i - j))) = 0$.*

*Proof.* The attention scores $(\mathbf{s}_{ij})$ are bounded above by some $S > 0$ (Chiang et al., 2023, Prop. 21). Recall that the smallest positive number in $\mathbb{F}$ is $2^{-s}$, so there is a score $s_0 = \log 2^{-s-1}$ such that $\mathrm{round}(\exp s_0) = 0$. Let $\Delta_a = (S - s_0)/a$. Then if $i - j \geq \Delta_a$, then $\mathrm{round}(\exp \mathbf{s}_{ij}) = 0$. □

**Proposition F.9.** *Let $T$ be a depth-$k$ fixed-precision transformer with ALiBi. There exists a depth-$k$ $\mathsf{TL}[\overline{\#}, \mathbf{Y}]$ formula $\phi_T$ that simulates $T$.*

*Proof sketch.* If $a = 0$, we can use the same construction as in Proposition B.6. If $a > 0$, we cannot use the trick of enumerating all possible queries, as in the proof of Proposition B.6. Instead, by Lemma F.8, there is a finite window $[i - \Delta_a, i]$ which receives nonzero attention. At query position $i$, we can use formulas $\phi_{\langle \mathbf{k}_c^{(\ell)} \rangle_b}, \mathbf{Y}^1 \phi_{\langle \mathbf{k}_c^{(\ell)} \rangle_b}, \ldots, \mathbf{Y}^{\Delta_a} \phi_{\langle \mathbf{k}_c^{(\ell)} \rangle_b}$ to obtain the keys at positions $i, i - 1, \ldots, i - \Delta_a$. We can then use Lemma B.5 to compute the attention scores according to Eq. (25). □

**Theorem F.10.** *A depth-$(k + 1)$ fixed-precision transformer with ALiBi can recognize $E_{k+1}$, but no depth-$k$ fixed-precision transformer with ALiBi can.*

*Proof.* Firstly, $E_{k+1}$ is definable by a fixed-precision transformer of depth $k + 1$ even without ALiBi. Secondly, by Proposition F.9, every language definable by a fixed-precision transformer with ALiBi is definable in $\mathsf{TL}[\overline{\#}, \mathbf{Y}]_k$, but by Theorem F.3, $E_{k+1}$ is not definable in $\mathsf{TL}[\overline{\#}, \mathbf{Y}]_k$. □

# G   Index of Notation

| | |
|---|---|
| $a$ | scaling factor for ALiBi (Appendix F.5) |
| $A, B$ | terms for numerator and denominator of attention (Proposition B.4) |
| $A_k, B_k$ | languages (Eq. (2)) |
| $\mathbf{a}, \mathbf{b}$ | numerator and denominator of sumdiv (Section 2.1) |
| $a, b$ | symbols ($\in \Sigma$) |
| $\alpha, \beta$ | formulas for numerator and denominator of attention (Proposition B.4) |
| $b$ | bit number ($\in [p]$, Proposition B.4) |
| $C$ | threshold in linear inequality (Appendix A.1 and Proposition B.4) |
| $C$ | transformation for counting (Theorem E.6) |
| $\mathbf{c}$ | attention output vector (Definition B.2) |
| $c$ | number of minimal depth-1 formulas (Appendix C.2 and Lemma D.1) |
| $c$ | coordinate ($\in [d]$, Proposition B.6) |
| $c$ | constant (Corollary E.8) |
| $D_k$ | language separating $\mathsf{TL}[\overleftarrow{\#}, \overrightarrow{\#}]$ depth levels (Theorem D.2) |
| $d$ | hidden dimension (Definition B.2) |
| $\Delta$ | window of positions $[i - \Delta]$ (Lemma F.8) |
| $e$ | neutral letter (Lemma F.2) |
| $E_k$ | $L_k$ with a neutral letter $e$ (Theorem F.3) |
| $E$ | word embedding (Definition B.2) |
| $\mathcal{F}$ | transformation (Theorem E.4) |
| $F$ | function $\Sigma^* \to \mathbb{F}^*$ (Lemma B.5) |
| $\mathbb{F}$ | all fixed-precision numbers (Definition B.1) |
| $f$ | function computing FFNN (Definition B.2) |
| $g$ | function $\mathbb{F} \to \mathbb{F}$ (Lemma B.5) |
| $h$ | $b$-intercept (Appendix C.2) |
| $\mathbf{h}$ | hidden vector (Definition B.2) |
| $\eta$ | learning rate (Section 5) |
| $I$ | interval (family of intervals) (Definition 2.5, Lemma 4.8, and Theorem 4.10) |
| $\mathcal{I}$ | set of all intervals (Definition 2.5, Lemma 4.8, and Theorem 4.10) |
| $\vec{i}, \vec{j}$ | endpoints of interval (Definition 2.5, Lemma 4.8, and Theorem 4.10) |
| $i, j$ | position ($\in [n]$) (Section 2 and Appendix A.1) |
| $\mathbf{K}, \mathbf{k}$ | key matrix/vector (Definition B.2) |
| $k$ | depth, number of blocks or pieces (Definitions A.2, B.2 and E.2) |
| $L$ | language ($\subseteq \Sigma^*$, Section 2) |
| $L_k$ | language (Eq. (1)) separating $\mathsf{TL}[\overleftarrow{\#}]$ depth levels (Theorem 4.10) |
| $\mathcal{L}$ | language recognized/defined by a formula or logic (Theorem E.7) |
| $\mathcal{L}$ | set of bodies of left-counting terms (Appendix C.1 and Lemmas 4.8 and 4.9) |
| $\ell$ | loop from 1 to $k - 1$ (Theorems D.2 and 4.10) |
| $\ell$ | index of half-plane or minimal depth-1 formula ($\in [c]$, Lemma 4.8) |
| $\ell$ | index of formula in $\widehat{\mathsf{MAJ}}$, term in a sum ($\in [m]$, Theorems E.4 and E.6) |
| $\ell$ | line (Lemma 4.8) |
| $\lambda$ | prefix family (Definition 4.1) |
| $\lambda$ | coefficient in linear inequality (Appendix A.1, Proposition B.4, and Lemma 4.9) |
| $\mathcal{M}$ | transformation (Theorem E.4) |
| $M$ | Parikh numerical predicate (Lemma 4.9) |
| $m$ | $|\Sigma|$, only used when $|\Sigma|$ would be circular (Definition 2.3) |
| $m$ | significand of fixed-precision number (Definition B.1) |

| | |
|---|---|
| $m$ | slope (Lemma 4.8) |
| $m$ | number of formulas in $\widehat{\mathrm{MAJ}}$, terms in a sum (Definition E.1 and Theorem E.4) |
| $n$ | length of $w$ (Section 2 and Appendix C.1) |
| $\vec{n}$ | Parikh vector of $w$ (Sections 2.3 and 4) |
| $\xi$ | truth assignment (Definition E.1) |
| $p$ | number of bits (Definition B.1) |
| $\Pi$ | PNP (Parikh numerical predicate) (Definition 2.6 and Lemma 4.9) |
| $\pi$ | function corresponding to PNP (Definition 2.6) |
| $\pi$ | permutation of $[n]$ (Appendix C.1) |
| $Q_\sigma$ | symbol predicate (Appendix A.1 and Definition E.1) |
| $\vec{q}$ | fixed-precision value of $\mathbf{q}$ (Proposition B.6) |
| $\mathbf{q}$ | query vector (Definition B.2) |
| $R$ | rotation matrix (Appendix F.3) |
| $\mathcal{R}$ | set of bodies of right-counting terms (Appendix C.1 and Lemmas D.1 and 4.9) |
| $\varrho$ | suffix family (Definition 4.1) |
| $\vec{s}$ | size of interval (Definition 4.3 and Lemma 4.8) |
| $s$ | number of fractional bits (Definition B.1) |
| $\Sigma$ | alphabet (Sections 2 and 2.3) |
| $\sigma$ | symbol ($\in \Sigma$) (Section 2.4) |
| $T$ | transformer (Section 3) |
| $\mathcal{T}$ | transformation (Theorem E.6) |
| $t$ | term (Appendix A.1) |
| $\mathbf{v}$ | value (Definition B.2) |
| $\vec{v}$ | Parikh vector (Section 2.3 and Lemma 4.8) |
| $W_{\mathrm{K}}, W_{\mathrm{Q}}, W_{\mathrm{V}}, W_{\mathrm{out}}$ | weight matrices (Definition B.2) |
| $w$ | string ($\in \Sigma^*$) (Section 2 and Definition 4.1) |
| $x$ | fixed-precision or real number (Section 3) |
| $x$ | padding size in reduction proof (Lemma F.2) |
| $x, y$ | formal variables (Definition E.1) |
| $y$ | index in reduction proof (Lemma F.2) |
| $\phi$ | formula (Section 4 and Appendix A.1) |
| $\chi$ | formula, esp. body of counting term (Lemma 4.8) |
| $\Psi$ | Parikh map ($\Sigma^* \to \mathbb{N}^{|\Sigma|}$) (Section 2.3) |
| $\psi$ | formula, esp. comparison (Lemmas 4.8 and 4.9 and Theorem E.6) |
| $\theta$ | angle for rotation matrix (Appendix F.4) |

