# OpenReview forum: "Knee-Deep in C-RASP: A Transformer Depth Hierarchy"
_NeurIPS.cc/2025/Conference — NeurIPS 2025 poster_

### Official Review · Reviewer_CcuD · 2025-06-08

**Clarity:** 3
**Significance:** 3
**Originality:** 3
**Rating:** 5
**Confidence:** 3

**Summary:**

This paper investigates the role of depth in transformer architectures by drawing a connection to a logical framework known as temporal logic with counting (TL[↼#]). The authors focus on fixed-precision transformers: all computations are performed with limited numerical precision, except within the attention mechanism. The core theoretical contribution is to show that these fixed-precision transformers are equivalent in expressivity to the programming language C-RASP, which is equivalent to the logic TL[↼#]. The equivalence is depth-preserving: the number of layers in the transformer corresponds to the nesting depth of the counting operators in the logic. Using this correspondence, the authors establish a depth hierarchy: deeper TL[↼#] formulas can define more complex sequence patterns than shallower ones. This implies that deeper transformers are more expressive than shallower ones. To support the theory, the paper presents empirical results showing that transformers require greater depth to solve formal language tasks as the complexity of the sequence dependencies increases. These results align with the theoretical depth requirements predicted by TL[↼#].

**Questions:**

1. Are there insights into whether width can compensate for depth (especially on these synthetic tasks), even if not in a fully expressive way?
2. The authors note performance degradation at higher k values (e.g., $k \geq 10$). Is this due to optimization difficulties, data sparsity, or theoretical limitations?

**Ethical Concerns:**

["NO or VERY MINOR ethics concerns only"]

**Final Justification:**

I am satisfied with the authors' response during rebuttal.

**Limitations:**

* The theoretical framework assumes fixed-precision computation and omits positional encodings. This diverges from common transformer use cases, which limits immediate applicability to current architectures.
* The experimental validation is limited to synthetic tasks. While effective for proving the point, these tasks do not capture the complexity or variability of natural language.
* While the authors prove depth separation for expressivity, they do not provide results regarding learnability or training stability at increasing depths.

**Paper Formatting Concerns:**

No concern.

**Quality:**

4

**Strengths And Weaknesses:**

**Strengths:**

* The authors provide a mathematically grounded depth hierarchy theorem, building on logics with counting and extending previous results in a novel and robust way.
* The equivalence between fixed-precision transformers and TL\[↼#] is not only elegant but also preserves depth, which is central to the paper’s claims.
* A careful experimental setup shows that the theory’s predictions about depth requirements hold up in practice, at least on synthetic language tasks.

**Weaknesses:**

* The results apply specifically to fixed-precision transformers without positional encodings. While this is reasonable in a theoretical sense, it limits immediate applicability to real-world transformer architectures.
* The empirical validation is limited to formal language tasks. It remains unclear how these results translate to real-world NLP tasks.
* The no-position-encoding assumption diverges from most practical models, which might limit the audience’s perception of the impact.

---

> ### Author Rebuttal · Authors · 2025-07-30
>
> Thank you very much for your review.
>
> > The results apply specifically to fixed-precision transformers without positional encodings. While this is reasonable in a theoretical sense, it limits immediate applicability to real-world transformer architectures....The no-position-encoding assumption diverges from most practical models, which might limit the audience’s perception of the impact.
>
> We acknowledge that this is a limitation of the paper in its present form. Although transformers with no position encodings (NoPE) have been considered in practice (Kazemnejad et al., 2023), we do have something to say about position encodings and would be happy to add it to the paper. It would be easy to extend Theorem 3.1 (fixed-precision transformers are equivalent to $\\mathsf{TL}[\\overset{\\leftharpoonup}{\\#}]$) to include periodic position encodings on the transformer side and modular predicates on the logic side. Then, by adding a neutral letter to $L_{k}$ (that is, by allowing some letter, say, $e$, to be inserted anywhere in a string without changing membership), we can show a strict depth hierarchy even in the presence of modular predicates, and therefore for transformers with periodic position encodings.
>
> > The empirical validation is limited to formal language tasks. It remains unclear how these results translate to real-world NLP tasks.
>
> It is true that our results are limited to formal language tasks, but the family of formal languages we use, $L_k$, may well be related to real-world tasks. Since it requires searching left to find an $a$, then searching left to find a $b$, and so on up to $k$ times, it can be seen as a prerequisite to any task that involves a chain of $k$ long-distance dependencies.
>
> For example, the $k$-hop induction heads task involves iterating the following procedure $k$ times: at position $i$, find a previous position $i'<i$ with the same symbol as $i$, then find a previous position $i''<i'$ with the same symbol as $i'+1$, and so on, which is essentially the same procedure needed to solve $L_k$. Although more work would be needed to make this more precise, we predict on the basis of our theory that $k$ hops require $k$ layers. (Sanford et al (2024) predict $\log k$ layers; the difference is that they assume a fixed string length, whereas we assume unbounded length.)
>
> Another basic application to NLP would be recognizing "stretched" $k$-grams. Our results imply that for any sequence of symbols $a_1, a_2, \ldots, a_k$, the language $\Sigma^* a_1^+ a_2^+ \cdots a_k^+ \Sigma^*$ requires depth $k$. This is immediately applicable to a modality where each symbol is "stretched" out, like recognizing words in speech. It would be an interesting question for future research to investigate the case without "stretching".
>
> > Are there insights into whether width can compensate for depth (especially on these synthetic tasks), even if not in a fully expressive way?
>
> Our results show that width cannot compensate for depth if we would like the transformer to operate uniformly over every input length. However, if we are satisfied with a fixed context window then it is possible. In fact, every finite language can be defined by a depth-2 $\\mathsf{TL}[\\overset{\\leftharpoonup}{\\#}]$ formula, which will imply the same for transformers. However, this requires a width exponential in the context window size. It would be an interesting question to study depth-width tradeoffs for $\\mathsf{TL}[\\overset{\\leftharpoonup}{\\#}]$ formulas in the fixed length regime.
>
> > The authors note performance degradation at higher k values (e.g., $k\ge10$). Is this due to optimization difficulties, data sparsity, or theoretical limitations?
>
> One possible cause is insufficient diversity in the training data. Because we restricted the training data to strings of length [101,150], languages $L_k$ for large $k$ tend to have very short contiguous blocks of $a$'s and $b$'s, in order to fit $k$ alternations within the string. Thus, the particular distribution of blocks may become an extraneous signal for $L_k$ that hampers learning as $k$ gets larger.
>
> Other possible causes could be: insufficient training time, insufficient width ($d$), or the so-called "curse of depth" (Sun et al., 2025).
>
> > While the authors prove depth separation for expressivity, they do not provide results regarding learnability or training stability at increasing depths.
>
> Our results sharply mark the boundary between expressible and inexpressible functions at varying depths, allowing future investigation to isolate issues of learnability and training stability. Two questions in this vein arise due to our results. First, why is there degradation in learnability past $k=10$, despite the languages being expressible? Second, why is there non-zero performance for $k$ which are inexpressible?
>
> Kazemnejad et al., 2023. The Impact of Positional Encoding on Length Generalization in Transformers. NeurIPS. arXiv:2305.19466.
>
> Sanford et al., 2024. Transformers, parallel computation, and
> logarithmic depth. ICML. arXiv:2402.09268.
>
> Sun et al., 2025. The Curse of Depth in Large Language Models. arXiv:2502.05795.

---

> > ### Comment · Reviewer_CcuD · 2025-08-07
> >
> > I thank the authors for the detailed rebuttal. I appreciate the clarifications regarding position encodings, connections to real-world tasks, and potential extensions of your theoretical results. Your responses addressed my main concerns and demonstrated both the relevance and future potential of the work. In particular, the proposed extensions to periodic position encodings and the discussion on stretched n-gram recognition helped clarify the broader applicability. I will raise my score.

---

> > > ### Author Response · Authors · 2025-08-07
> > >
> > > Thank you very much! We appreciate your feedback, which will definitely improve the paper.

---

> > > > ### Author Response · Authors · 2025-08-08
> > > > **Supplementary Information Regarding Experiments**
> > > >
> > > > We would like to update those reviewers who were interested in the unusually low performance on learning $L_{10}$, even when the network had enough depth to express the langauge. We confirmed that increasing the data from 2000 to 10000 examples, increasing the training length from 150 to 350, and increasing the number of training epochs from 20 to 100 enabled the depth 8 and 9 models to learn $L_{10}$ to a high accuracy, and generalize beyond the lengths seen during training. Under the same constraints, the 7 layer transformer could not learn $L_{10}$ in a way that length-generalized. We also found, tentatively, that increasing width (d) and layer-norm scaling (Sun et al., 2005) did not help.
> > > >
> > > >
> > > > 7 layer evaluation results:
> > > > - Accuracy on 301_to_350: 97.79%
> > > > - Accuracy on 401_to_450: 85.10%
> > > > - Accuracy on 501_to_550: 48.03%
> > > > - Accuracy on 601_to_650: 18.80%
> > > >
> > > > 8 layer evaluation results:
> > > > - Accuracy on 301_to_350: 99.94%
> > > > - Accuracy on 401_to_450: 99.54%
> > > > - Accuracy on 501_to_550: 98.11%
> > > > - Accuracy on 601_to_650: 88.11%
> > > >
> > > > 9 layer evaluation results:
> > > > - Accuracy on 301_to_350: 99.98%
> > > > - Accuracy on 401_to_450: 99.17%
> > > > - Accuracy on 501_to_550: 90.22%
> > > > - Accuracy on 601_to_650: 70.99%
> > > >
> > > >
> > > > Sun et al., 2025. The Curse of Depth in Large Language Models. arXiv:2502.05795

---

### Official Review · Reviewer_AK8r · 2025-06-27

**Clarity:** 2
**Significance:** 3
**Originality:** 3
**Rating:** 4
**Confidence:** 3

**Summary:**

This paper uses a simplified form of a transformer to study the relation between depth of the transformer (number of stacked transformer blocks) and depth in a formal language modeled by TL (temporal Logic) which is equivalent to C-RASP. The paper establishes a tight connection between depth of a transformer model and a programming language: C-RASP. Using the established connection, authors claim to characterize a family of transformers that can length-generalize.

**Questions:**

1. In empirical results, authors claim that the drop after 10 layers does not contradict their findings. I thought their findings were tight so I am surprised why this happens. Can you elaborate?

2. Can you elaborate on how findings on TL generalize to other classes of logic?

3. It is unclear to me if the choice of no positional embedding is to ease theoretical demonstrations or if it is something that also helps improve results. Having no positional embeddings can clearly be problematic in many cases. It would help to have a more clear explanation on this.

**Ethical Concerns:**

["NO or VERY MINOR ethics concerns only"]

**Final Justification:**

It's a good paper with string results, but presentation is rough and there are some strong assumption / surprises in the paper.

**Limitations:**

The paper's language highly relies on notations and definitions defined in a narrow set of papers. It may be hard to follow for a general audience. Empirical results are limited and there is a gap there which is uncovered (see Question 1 above)

**Quality:**

4

**Strengths And Weaknesses:**

Strengths: The paper presents a solid theoretical work, and proves a non-trivial result.

Weaknesses:

The paper accumulates a large number of definitions and also heavily relies on objects defined in a few previous background papers (e.g. Chiang et al 2024) which are necessary for readers to follow the paper's arguments. It is therefore a hard to follow paper. Authors could make an effort in summarizing the background content required for the reader to read through the paper and only present the necessary.

It is also not clear which of the assumptions are made to make proof of theoretical results easier and which are made to improve the model / results.

Empirical experiments have some shortcomings. Authors seem to have run some more experiments during the rebuttal. I am uncomfortable with those results since they seem to have let the model train for longer / have slightly different hyperparameters.

---

> ### Author Rebuttal · Authors · 2025-07-30
>
> Thank you very much for your review.
>
> > The paper accumulates a large number of definitions and also heavily relies on objects defined in a few previous background papers (e.g. Chiang et al 2024) which are necessary for readers to follow the paper's arguments.
>
> We agree that the paper is heavy on formalism, and think that Reviewer CRAK's suggestion to add examples of C-RASP and/or $\\mathsf{TL}[\\overset{\\leftharpoonup}{\\#}]$ would be helpful. We will do so in the final version, either in the extra page or in the appendix.
>
> We also agree that there are quite a few definitions used in the proof of Section 4 (accommodating, Parikh numerical predicates, etc.). These definitions represent our effort to clarify ideas developed in previous papers (esp. Behle et al., 2009). We will put some further effort into making the argument clearer, especially the sketch in Section 4.1.
>
> > 1. In empirical results, authors claim that the drop after 10 layers does not contradict their findings. I thought their findings were tight so I am surprised why this happens. Can you elaborate?
>
> One possible cause is insufficient diversity in the training data. Because we restricted the training data to strings of length [101,150], languages $L_k$ for large $k$ tend to have very short contiguous blocks of $a$'s and $b$'s, in order to fit $k$ alternations within the string. Thus, the particular distribution of blocks may become an extraneous signal for $L_k$ that hampers learning as $k$ gets larger.
>
> Other possible causes could be: insufficient training time, insufficient width ($d$), or the so-called "curse of depth" (Sun et al., 2025).
>
> > 2. Can you elaborate on how findings on TL generalize to other classes of logic?
>
> In Appendix E we derive a depth hierarchy for $\\widehat{\\mathsf{MAJ}}_2[<]$, a logic studied by Behle et al. (2009). Our results can also be extended straightforwardly to show a depth hierarchy for the extension of $\\mathsf{TL}[\\overset{\\leftharpoonup}{\\#}]$ with successor and modular predicates. An interesting future question is whether this can be shown for the logic with arbitrary numerical predicates, which would correspond to $\\mathsf{TC}^0$ circuits with linear gates.
>
> Behle et al., 2009. Regular Languages Definable by Majority Quantifiers with Two Variables. International Conference on Developments in Language Theory.
>
> Sun et al., 2025. The Curse of Depth in Large Language Models. arXiv:2502.05795.

---

> > ### Comment · Reviewer_AK8r · 2025-08-08
> >
> > thank you. Can you explain why you do not use positional encoding in your models?

---

### Official Review · Reviewer_zNc9 · 2025-07-02

**Clarity:** 3
**Significance:** 3
**Originality:** 2
**Rating:** 5
**Confidence:** 2

**Summary:**

Building on prior work in C-RASP and temporal logic (TL[#]), the authors show that (a subclass of) transformers are equivalent to C-RASP and TL[#]-left. This subclass includes models where computations use fixed-precision rounding everywhere except within the attention mechanism. Based on this equivalence, the paper establishes a "depth hierarchy," demonstrating that there are specific tasks deeper transformers can solve which shallower ones cannot. Finally, the authors provide empirical evidence showing their theoretical predictions on model expressivity align well with practice. They test this on a next-token prediction variant of the $L_k$ language recognition task.

**Questions:**

1.  The work of Yang and Chiang (2024) seems to present similar results regarding the equivalence of C-RASP, TL[#]-left, and certain transformers. Could you elaborate on how your work differs from theirs? In particular, their work was not included in the comparison table in Section 1, and the equivalence of transformers and TL[#]-left, which appears to be their contribution, is presented as a contribution of the current work on line 29. Clarifying the distinction would be very helpful.

2.  Based on the equivalence results you provide, could you elaborate on the depth required to learn tasks like parity and modular addition?

3. Regarding the experiments, accuracy is shown to sharply decline for $k \geq 10$, even for models with sufficient depth. You attribute this to challenges with learnability, not expressibility. Could you provide more insight into the specific learning challenges for these more difficult cases and speculate on what might be required to solve them?

**Ethical Concerns:**

["NO or VERY MINOR ethics concerns only"]

**Final Justification:**

I recommend accepting the paper. It is well-written and clearly presented, with a strong theoretical contribution demonstrating a clear depth hierarchy under non-prohibitive constraints. The inclusion of empirical results to support the theoretical findings further strengthens the work by highlighting its practical applicability. While the paper could be improved by adding a brief introduction to temporal logic and C-RASP to enhance accessibility, these are minor issues that do not detract from the overall quality and significance of the contribution.

**Limitations:**

yes

**Quality:**

4

**Strengths And Weaknesses:**

**Strength:**
The paper is well-written, and the presentation is solid. The theoretical demonstration of a clear depth hierarchy under non-prohibitive constraints is a very interesting contribution to the broader community. I also appreciate that the authors provided empirical results to support their theoretical findings. This strongly demonstrates the practical applicability of their results and shows that the underlying theoretical assumptions were practical.

**Areas for Improvement:**
The paper's primary weakness is its accessibility. While the results are interesting for a general audience, the work may be difficult to follow for those not already familiar with temporal logic or C-RASP. I understand that these concepts are built on previous work, but including more intuitive or illustrative examples of them would be very helpful. Adding a brief introductory section on these topics in an appendix could significantly improve the paper's clarity and open it up to a larger audience.

---

> ### Author Rebuttal · Authors · 2025-07-30
>
> Thank you very much for your review.
>
> > [I]ncluding more intuitive or illustrative examples of [temporal logic and C-RASP] would be very helpful. Adding a brief introductory section on these topics in an appendix could significantly improve the paper's clarity and open it up to a larger audience.
>
> We agree that examples would be helpful, and will add examples in the final version, either in the extra page or in the appendix.
>
> > 1. The work of Yang and Chiang (2024) seems to present similar results regarding the equivalence of C-RASP, TL[#]-left, and certain transformers. Could you elaborate on how your work differs from theirs? In particular, their work was not included in the comparison table in Section 1, and the equivalence of transformers and TL[#]-left, which appears to be their contribution, is presented as a contribution of the current work on line 29. Clarifying the distinction would be very helpful.
>
> Yang and Chiang (2024) did not prove an exact equivalence; they proved that $\\mathsf{TL}[\\overset{\\leftharpoonup}{\\#}]$ can be converted to an arbitrary-precision transformer, and a transformer with $O(1)$ precision (even inside attention) can be converted to $\\mathsf{TL}[\\overset{\\leftharpoonup}{\\#}]$. Here, we have tightened these bounds by proving that transformers with $O(1)$ precision (except inside attention) are exactly equivalent to $\\mathsf{TL}[\\overset{\\leftharpoonup}{\\#}]$. We will clarify these differences in the final version.
>
> (The comparison table at the end of Section 1 is for depth separation results, and Yang and Chiang (2024) did not prove any depth separations, so their paper is not included in the table.)
>
> > 2. Based on the equivalence results you provide, could you elaborate on the depth required to learn tasks like parity and modular addition?
>
> Our equivalence between transformers and C-RASP proves that tasks like parity and modular addition cannot be expressed by transformers of any depth (without positional encodings). Huang et al. (2025) show that C-RASP programs cannot perform modular counting, which is a prerequisite for both parity and modular addition.
>
> > 3. Regarding the experiments, accuracy is shown to sharply decline for, even for models with sufficient depth. You attribute this to challenges with learnability, not expressibility. Could you provide more insight into the specific learning challenges for these more difficult cases and speculate on what might be required to solve them?
>
> One possible cause is insufficient diversity in the training data. Because we restricted the training data to strings of length [101,150], languages $L_k$ for large $k$ tend to have very short contiguous blocks of $a$'s and $b$'s, in order to fit $k$ alternations within the string. Thus, the particular distribution of blocks may become an extraneous signal for $L_k$ that hampers learning as $k$ gets larger.
>
> Other possible causes could be: insufficient training time, insufficient width ($d$), or the so-called "curse of depth" (Sun et al., 2025).
>
> Huang et al., 2025. A Formal Framework for Understanding Length Generalization in Transformers. ICLR. arXiv:2410.02140.
>
> Sun et al., 2025. The Curse of Depth in Large Language Models. arXiv:2502.05795.
>
> Yang and Chiang, 2024. Counting Like Transformers: Compiling Temporal Counting Logic Into Softmax Transformers. CoLM. arXiv:2404.04393.

---

> ### Comment · Reviewer_zNc9 · 2025-08-01
>
> Thank you for your response.
>
> > Tasks like parity and modular addition cannot be expressed by transformers of any depth.
>
> Could you please clarify whether "expressed" in this context refers to the ability to recognize the language (or solve the task) for any inputs length, or successful length generalization? Many of these tasks can be learned in-distribution, meaning for the input lengths seen during training. Does the framework offer any insights into in-distribution learning, even if it does not address generalization to longer sequences?
>
> > languages $L_k$ for large $k$ tend to have very short contiguous blocks of $a$'s and $b$'s, in order to fit $k$ alternations within the string
>
> There appears to be a sudden performance drop from $L_8$ to $L_9$, despite the fact that the lengths of contiguous blocks are very close in these cases. This is a notable artifact in the figure, and the experimental results could be further strengthened if the authors investigate it more closely, potentially by increasing training time or model width, as suggested by the authors.

---

> > ### Author Response · Authors · 2025-08-04
> > **Response to Reviewer**
> >
> > Thank you for the response, see our replies below:
> >
> > > Could you please clarify whether "expressed" in this context refers to the ability to recognize the language (or solve the task) for any inputs length, or successful length generalization? Many of these tasks can be learned in-distribution, meaning for the input lengths seen during training. Does the framework offer any insights into in-distribution learning, even if it does not address generalization to longer sequences?
> >
> > Our results are about expressivity over unbounded input lengths. Towards your question, this means that we claim fixed-precision transformers without positional encodings cannot express a solution to modular addition or parity that will always work out-of-distribution (length-generalization). Our framework suggests that a model may need a size exponential in the input length in order to express functions like parity or modular addition in the in-distribution setting (where we only consider inputs up to a bounded context window length).
> >
> > > There appears to be a sudden performance drop from $L_8$ to $L_9$, despite the fact that the lengths of contiguous blocks are very close in these cases. This is a notable artifact in the figure, and the experimental results could be further strengthened if the authors investigate it more closely, potentially by increasing training time or model width, as suggested by the authors.
> >
> > We thank the reviewer for the suggestion and will prioritize these experiments for an updated draft of the paper.

---

> > > ### Comment · Reviewer_zNc9 · 2025-08-05
> > >
> > > Thank you for your clarification. I have no further questions and wish the authors the best of luck!

---

### Official Review · Reviewer_CRAK · 2025-07-05

**Clarity:** 3
**Significance:** 3
**Originality:** 3
**Rating:** 4
**Confidence:** 2

**Summary:**

This paper aims to investigate the expressive power of a specific transformer, the future-masked fixed-precision transformer. The authors first establish an equivalence between this transformer and the programming language C-RASP, then prove a depth hierarchy for the language, demonstrating that deeper C-RASP models possess greater expressive power. Overall, the paper provides a clear understanding of how the expressive power of transformers varies with depth. The accompanying experiments are well aligned with the theoretical analysis.

**Questions:**

In addition to the concern raised in the Weaknesses section, I have a question regarding the experiments. In some cases, the accuracy for the depth-(k-1) model remains quite high, while for depth-(k-2) and lower, the accuracy drops significantly. For example, in Figure 5, the "Accuracy on [201,250]" shows this. Why does the depth-(k-1) model perform so differently compared to models with lower depth? Could you provide some explanation or intuition for this observation?

**Ethical Concerns:**

["NO or VERY MINOR ethics concerns only"]

**Limitations:**

As noted in the Weaknesses and Questions sections, the omission of residual connections and positional encoding may impact the generality of the results. I am not very familiar with this research area, while the statement seems valid with sufficient experiemtns, I lean toward accepting this paper, if there are any misunderstandings in my comments, please let me know.

**Quality:**

3

**Strengths And Weaknesses:**

Strengths

1. This work establishes a connection between transformers of varying depth and a specially designed language with corresponding depth, making it possible to analyze and understand the expressive power of these models.

2. The paper rigorously analyzes the expressive power of transformers with varying depths, accomply with solid theoretical proofs, provding insights for mechanisms interpretion and architecture design for practical transformers.

3. The experimental results are consistent with the theoretical findings.

Weaknesses

The analysis appears to overlook an important aspect of practical transformers: the residual connections between layers. Additionally, since positional encoding is not considered in this work, it may be challenging for the model to distinguish tokens at different positions, potentially limiting its expressive power.

---

> ### Author Rebuttal · Authors · 2025-07-30
>
> Thank you very much for your review.
>
> > The analysis appears to overlook an important aspect of practical transformers: the residual connections between layers.
>
> Sorry for the mistake. We did assume residual connections, and our translation from $\\mathsf{TL}[\\overset{\\leftharpoonup}{\\#}]$ to transformers depends on them.
>
> Definition A.8 does not specify that $f$ has to be a FFNN, so $f$ could include a residual connection.
>
> However, equation 8 indeed omitted residual connections for self-attention layers. In the final version, we will correct the equation to $\\mathbf{h}^{(\\ell)}\_{i}(w) = f^{(\\ell)}\left(\\mathbf{c}^{(\\ell)}\_{i}(w) + \\mathbf{h}^{(\\ell-1)}_i(w)\\right)$.
>
> > Additionally, since positional encoding is not considered in this work, it may be challenging for the model to distinguish tokens at different positions, potentially limiting its expressive power.
>
> We acknowledge that this is a limitation of the paper in its present form. Although transformers with no position encodings (NoPE) have been considered in practice (Kazemnejad et al., 2023), we do have something to say about position encodings and would be happy to add it to the paper. It would be easy to extend Theorem 3.1 (fixed-precision transformers are equivalent to $\\mathsf{TL}[\\overset{\\leftharpoonup}{\\#}]$) to include periodic position encodings on the transformer side and modular predicates on the logic side. Then, by adding a neutral letter to $L\_{k}$ (that is, by allowing some letter, say, $e$, to be inserted anywhere in a string without changing membership), we can show a strict depth hierarchy even in the presence of modular predicates, and therefore for transformers with periodic position encodings.
>
> > In some cases, the accuracy for the depth-(k-1) model remains quite high, while for depth-(k-2) and lower, the accuracy drops significantly. For example, in Figure 5, the "Accuracy on [201,250]" shows this. Why does the depth-(k-1) model perform so differently compared to models with lower depth? Could you provide some explanation or intuition for this observation?
>
> If we understand correctly, you are asking why the numbers in the diagonal just below the the black line are considerably higher than the next diagonal below. In general, the presence of non-zero numbers below the line can be explained by the fact that the training and test data consist of strings of bounded length, whereas the theory only makes guarantees about strings of unbounded length. It stands to reason that these non-zero numbers would fade the further one goes below the line, and fade more sharply as the test length increases. But whether there is something special about the diagonal immediately below the line is not so clear to us, and we don't have any guesses about what would cause that if true.
>
> Kazemnejad et al., 2023. The Impact of Positional Encoding on Length Generalization in Transformers. NeurIPS. arXiv:2305.19466.

---

> > ### Comment · Reviewer_CRAK · 2025-08-02
> >
> > Thanks for the clarification. I will keep my score as I lean towards accepting this paper.

---

### Decision · Program_Chairs · 2025-09-17

**Decision:**

Accept (poster)

**Comment:**

(a) Summary

The paper proves a depth-preserving equivalence between fixed-precision transformers (with residuals, no positional encodings) and C-RASP, then establishes a strict depth hierarchy - deeper models are strictly more expressive. Experiments on formal language tasks support the theory, and additional runs during rebuttal confirm that models below the predicted depth fail to length-generalize.

(b) Strengths

- Strong theoretical contribution with a clear depth hierarchy result.
- Tightens prior work by proving exact equivalence.
- Empirical evidence aligns with theory, rebuttal adds convincing results.
- Provides a framework for reasoning about depth and generalization.

(c) Weaknesses

- Assumes no positional encodings.
- Maybe hard to follow for non-specialists.
- Experiments are synthetic, real-world implications are discussed but not shown.

(d) Reason

Even with some weaknesses, the theoretical advance is significant and well-supported.

(e) Discussion & Rebuttal

Reviewers raised concerns about residuals, positional encodings, and empirical anomalies. Authors clarified residuals were assumed, outlined how to handle PEs, and ran stronger experiments confirming the depth threshold. Concerns about accessibility remain, but technical soundness and novelty are clear.